# Mouse models of human *PIK3CA*-related brain overgrowth have acutely treatable epilepsy

Achira Roy[1], Jonathan Skibo[1], Franck Kalume[1], Jing Ni[2], Sherri Rankin[3], Yiling Lu[4], William B Dobyns[1], Gordon B Mills[4], Jean J Zhao[2], Suzanne J Baker[3], Kathleen J Millen[1]*

[1]Center for Integrative Brain Research, Seattle Children's Research Institute, Seattle, United States; [2]Department of Cancer Biology, Dana Farber Cancer Institute, Boston, United States; [3]Department of Developmental Neurobiology, St. Jude Children's Research Hospital, Memphis, United States; [4]The University of Texas MD Anderson Cancer Center, Houston, United States

**Abstract** Mutations in the catalytic subunit of phosphoinositide 3-kinase (*PIK3CA)* and other PI3K-AKT pathway components have been associated with cancer and a wide spectrum of brain and body overgrowth. In the brain, the phenotypic spectrum of *PIK3CA*-related segmental overgrowth includes bilateral dysplastic megalencephaly, hemimegalencephaly and focal cortical dysplasia, the most common cause of intractable pediatric epilepsy. We generated mouse models expressing the most common activating *Pik3ca* mutations (*H1047R* and *E545K*) in developing neural progenitors. These accurately recapitulate all the key human pathological features including brain enlargement, cortical malformation, hydrocephalus and epilepsy, with phenotypic severity dependent on the mutant allele and its time of activation. Underlying mechanisms include increased proliferation, cell size and altered white matter. Notably, we demonstrate that acute 1 hr-suppression of PI3K signaling despite the ongoing presence of dysplasia has dramatic anti-epileptic benefit. Thus PI3K inhibitors offer a promising new avenue for effective anti-epileptic therapy for intractable pediatric epilepsy patients.

*For correspondence: kathleen. millen@seattlechildrens.org

**Competing interests:** The authors declare that no competing interests exist.

## Introduction

The phosphoinositide-3 kinase (PI3K)-AKT pathway is a central player of intracellular signaling, conserved from yeast to mammals. Activating mutations in genes of PI3K-AKT signaling pathway, especially *PIK3CA*, encoding the catalytic p110α isoform of the PI3K complex, have long been linked to cancer (*Cheung and Testa, 2013*; *Engelman, 2009*; *Hennessy et al., 2005*; *Iwabuchi et al., 1995*; *Samuels and Waldman, 2011*; *Gymnopoulos et al., 2007*). Germline and mosaic mutations of *PIK3CA* and other pathway genes also cause a wide range of brain and body overgrowth disorders; all anomalies caused by somatic *PIK3CA* mutations are now collectively termed *PIK3CA*-Related Overgrowth Spectrum (PROS) (*Keppler-Noreuil et al., 2014*). The broad spectrum of brain overgrowth disorders caused by *PIK3CA* activating mutations is impressive. Three strongly activating *PIK3CA* mutations found most commonly in cancer (hotspot mutations) result in severe segmental cortical dysplasia (SEGCD), which includes bilateral dysplastic megalencephaly (MEG), hemimegalencephaly (HMEG) and focal cortical dysplasia (FCD) types 2a/2b (*Lee et al., 2012*; *D'Gama et al., 2015*; *Conway et al., 2007*; *Jansen et al., 2015*). Other mutations, resulting in intermediate or weak *PIK3CA* activation, cause MEG or MEG with polymicrogyria (MEG-PMG) as part of the MEG-capillary malformation syndrome (MCAP) (*Conway et al., 2007*; *Mirzaa et al., 2012*; *Rivière et al.,*

**eLife digest** An enzyme called PI3K is involved in a major signaling pathway that controls cell growth. Mutations in this pathway have devastating consequences. When such mutations happen in adults, they can lead to cancer. Mutations that occur in embryos can cause major developmental birth defects, including abnormally large brains. After birth, these developmental problems can cause intellectual disabilities, autism and epilepsy. Children with this kind of epilepsy often do not respond to currently available seizure medications.

There are several outstanding questions that if answered could help efforts to develop treatments for children with brain growth disorders. Firstly, how do the developmental abnormalities happen? Do the abnormalities themselves cause epilepsy? And can drugs that target this pathway, and are already in clinical trials for cancer, control seizures?

Now, Roy et al. have made mouse models of these human developmental brain disorders and used them to answer these questions. The mice were genetically engineered to have various mutations in the gene that encodes the catalytic subunit of the PI3K enzyme. The mutations were the same as those found in people with brain overgrowth disorders, and were activated only in the developing brain of the mice. These mutations caused enlarged brain size, fluid accumulation in the brain, brain malformations and epilepsy in developing mice – thus mimicking the human birth defects. The severity of these symptoms depended on the specific mutation and when the mutant genes were turned on during development.

Next, Roy et al. studied these mice to see if the seizures could be treated using a drug, that has already been developed for brain cancer. This drug specifically targets and reduces the activity of PI3K. Adult mutant mice with brain malformations were treated for just one hour; this dramatically reduced their seizures. These experiments prove that seizures associated with this kind of brain overgrowth disorder are driven by ongoing abnormal PI3K activity and can be treated even when underlying brain abnormalities persist. Roy et al. suggest that drugs targeting PI3K might help treat seizures in children with these brain overgrowth disorders.

*2012*). Developmental features of these brain disorders include cortical malformations, hydrocephalus, Chiari malformation, intellectual disability, autism and epilepsy (*Keppler-Noreuil et al., 2014*; *Mirzaa et al., 2012*). FCD represents one of the most common causes of intractable epilepsy (*Bast et al., 2006*; *Fauser et al., 2015*; *Fauser, 2006*).

Conditional mouse alleles for the *H1047R* and *E545K Pik3ca* hotspot mutations have been generated to study tumor formation and assess anti-cancer activities of pathway inhibitors (*Kinross et al., 2012*; *Liu et al., 2011*; *Meyer et al., 2011*; *Robinson et al., 2012*; *Yuan et al., 2013*). To understand the cellular mechanisms behind *PIK3CA*-related brain overgrowth disorders, we used a series of *cre*-drivers to activate expression of *H1047R* and *E545K* alleles in subsets of neural progenitors. Dramatic phenotypes resulted, faithfully modeling the entire spectrum of *PIK3CA*-associated human brain disorders including enlarged brain size, hydrocephalus, cortical dysplasia and epilepsy. The severity of these brain phenotypes critically depended on the *Pik3ca* allele and its time of activation. Notably, *Pik3ca*-associated epilepsy in mice was independent of brain overgrowth and cortical dysplasia. Further the seizures of adult megalencephalic mice were suppressed by acute 1 hr-administration of pan-PI3K inhibitor BKM120 (*Maira et al., 2012*). We conclude that epilepsy in these models represents an active *Pik3ca*-driven process that can be restricted by dynamic modulation of PI3K pathway activity in dysmorphic brains. These results raise the exciting prospect of new molecular based epilepsy therapies in patients whose seizures have been intractable to current anti-seizure therapies.

## Results

### Megalencephaly caused by Pik3ca overactivation is dependent on both the nature of the mutant allele and its time of overactivation

Two conditional *Pik3ca* activating alleles (*H1047R* and *E545K)* were crossed with *cre*-drivers to over-activate p110α in progressively restricted sets of neural progenitors and their progeny. The broadest distribution was achieved with *Nestin-cre*, being expressed in nearly all neural progenitors from early embryonic stages. A subset of late embryonic progenitors was targeted by *hGFAP-cre*; while tamoxifen-inducible *Nestin-creER* line drove *cre*-expression in a small subset of neural progenitors around birth. Expression of *H1047R* was dependent upon a tri-allelic system with tet-inducible mutant human cDNA activated by cre-dependent expression of the tet-activator protein (*Liu et al., 2011*) (*Figure 1—figure supplement 1*). The *E545K* mutation was knocked into the endogenous *Pik3ca* locus and a lox-stop-lox cassette introduced upstream of the initiation-coding exon, rendering the mutant allele cre-dependent (*Robinson et al., 2012*). The activity of all cre drivers was confirmed using reporter lines (*Figure 1—figure supplement 2*).

The most severe phenotype was achieved in *hGFAP-cre;H1047R* mutants, when doxycycline was administered from embryonic day (E)0.5. All mutants exhibited progressive hydrocephalus and died prior to weaning. Hydrocephalus was evident as a domed forehead at postnatal day (P)21 (*Figure 1b*). Hematoxylin-eosin stained P3 sections showed ventriculomegaly in the megalencephalic *H1047R* mutant brains. Strikingly the hippocampus was not evident in these mutants. Instead, the medial tissue was highly dysplastic with multiple infoldings along its entire length (*Figure 1c,d*). In contrast, when pups were treated with doxycycline from P1, no morphological differences were observed between the control and the *hGFAP-cre;H1047R* mutant (*Figure 1—figure supplement 3*). Thus the effect of *H1047R* mutation on brain size was dependent on time of activation.

*E545K* mice with the same *cre*-driver (*hGFAP-cre;E545K)* had a milder phenotype, surviving as adults without hydrocephalus, though their brain size was significantly larger compared to control littermates (*Figure 1e,h,j*). This provides evidence that with identical time of activation by the same cre driver, the brain phenotypes are dependent on specific *Pik3ca* allele. Earlier activation of *E545K* mutation with *Nestin-cre* led to an even more striking 54.4% volumetric increase, with mild ventriculomegaly and no hydrocephalus (*Figure 1f,h,k*). Interestingly, neonatal activation of *E545K* using *Nestin-creER* had no apparent impact on brain size (*Figure 1g,h,l*). Enlarged head size in both *Nestin-cre;E545K* and *hGFAP-cre;E545K* mutants was evident at birth and brain size of all the three adult *E545K* mutants was relatively stable (data not shown). Unlike *H1047R* mutants, gross morphology was normal for all *E545K* mutants. We conclude that brain phenotypes caused by Pik3ca-overactivation are both allele and time dependent. Further, we conclude that to cause brain overgrowth, over-activation of Pik3ca function is required during embryogenesis.

### Multiple allele-dependent embryonic mechanisms drive *Pik3ca*-MEG

To assess the mechanisms causing Pik3ca-dependent embryonic brain enlargement, we focused our analysis on *hGFAP-cre;H1047R* (doxycycline from E0.5) and *Nestin-cre;E545K* mutants, since these allelic combinations had the most extreme megalencephalic phenotypes.

The inner cortical length of *hGFAP-cre;H1047R* mutants was longer than controls at both E14.5 (p<0.01) and E16.5 (p<0.001; *Figure 2b–g*). This was accompanied by enlarged nuclear and cell size at both ages and decreased cell density at E16.5, but not increased proliferation or cell cycle exit (*Figure 3c–l*). Total cell number per cortical column length was not significantly different between control and *H1047R* mutant both at E14.5 and E16.5. Also, TUNEL$^+$ cell number was significantly lower in E16.5 mutant cortex than in control (p<0.01), indicating reduced apoptosis (*Figure 3—figure supplement 1a,c*); however the overall TUNEL$^+$ cell numbers for both control and mutant were small. Together these results indicate that cortical expansion in *hGFAP-cre;H1047R* mutant was not primarily driven by increased proliferation or reduced apoptosis; rather reduced cell density and increased cell size during embryogenesis were major contributing factors.

By contrast, in *Nestin-cre;E545K* mutants, the inner cortical length was comparable to controls at E14.5 but elongated at E16.5 (p<0.01); cortical thickness was slightly reduced in E16.5 (p<0.05) as compared to controls (*Figure 2h–m*). The labeling index was similar to the control at E14.5 but increased in E16.5 mutants (p<0.05), indicating more proliferation (*Figure 3m,o,r*). Cell density in

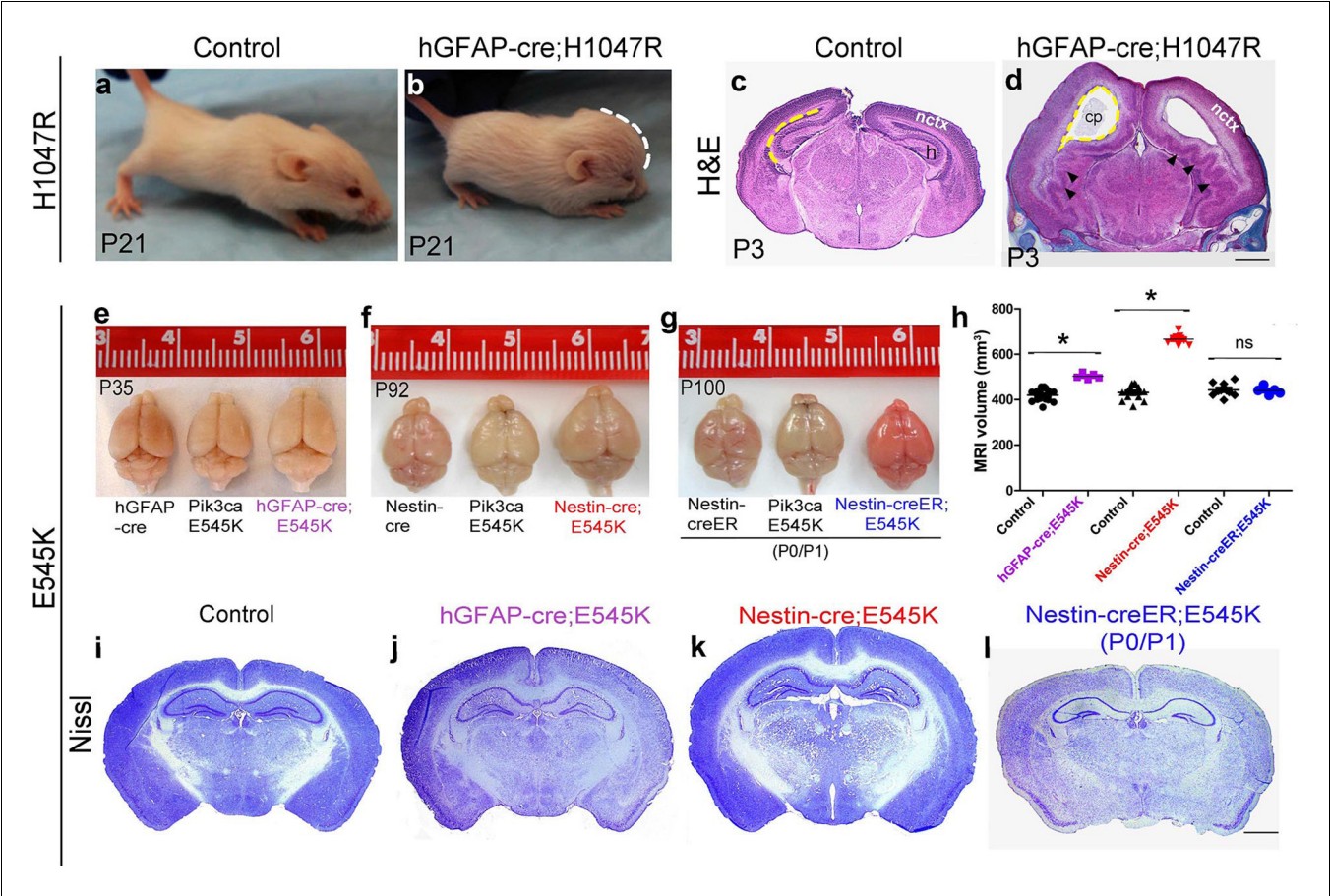

**Figure 1.** Embryonic *Pik3ca* overactivation in mice causes MEG. (**a,b**) Compared to control, P21 *hGFAP-cre;H1047R* mutants had domed foreheads. (**c, d**) Coronal section of H&E-stained P3 *H1047R* mutant showed bigger brain and enlarged lateral ventricles compared to control. Mutant neocortex (nctx) was dysplastic and medial tissue highly infolded (arrowhead; d). (**e–g**) P35 *hGFAP-cre;E545K* and *Nestin-cre;E545K* brains were noticeably larger than controls, while *Nestin-creER;E545K* mutants had normal-sized brains compared to controls. Red color of *Nestin-creER;E545K* brain is due to presence of a lox-stop-lox-Tomato reporter allele, and shows successful induction of cre activity. Controls for e,f and g are of genotypes *Pik3ca E545K, hGFAP-cre, Nestin-cre* and *Nestin-creER*. (**h**) MRI volumetric analyses of mutant and corresponding control brains. *p<0.0001; ns, not significant. Each data point in the graph represents 1 mouse. (**i–l**) Nissl-stained coronal sections of representative control and mutant brains. Scale bars: 1 mm (c,d); 2 mm (i-l). See also *Figure 1—figure supplements 1–3*.

The following figure supplements are available for figure 1:

**Figure supplement 1.** Genetic strategy for *Pik3ca* mouse models.

**Figure supplement 2.** Expression of *cre* lines.

**Figure supplement 3.** Neonatal activation of *H1047R* mutation show no effect on brain morphology

*E545K* mutant neocortex was similar to the control at E14.5 but was reduced at E16.5 (p<0.05). Total cells per cortical column length did not change in the *E545K* mutant; but the TUNEL$^+$ cell number was lower in E16.5 mutant cortex than in control (p<0.001). Intriguingly, the nuclear size of these mutant cells was similar to controls at both E14.5 and E16.5 but cell somas were significantly larger (p<0.05) at E16.5 (*Figure 3n,p,q,s,t*). The quit fraction indicative of rate of cells exiting cell cycle was significantly higher (p<0.05) in the *E545K* mutant between E15.5 and E16.5 (*Figure 3u,v*). At P35, the neocortical cells of *Nestin-cre;E545K* mutant were still larger (p<0.05) compared to controls (*Figure 3—figure supplement 2a,b*). Notably, in the adult P35 *Nestin-creER;E545K* mutant mice, where activation was initiated in neonates, and brain size was not different from controls, *E545K*-activated

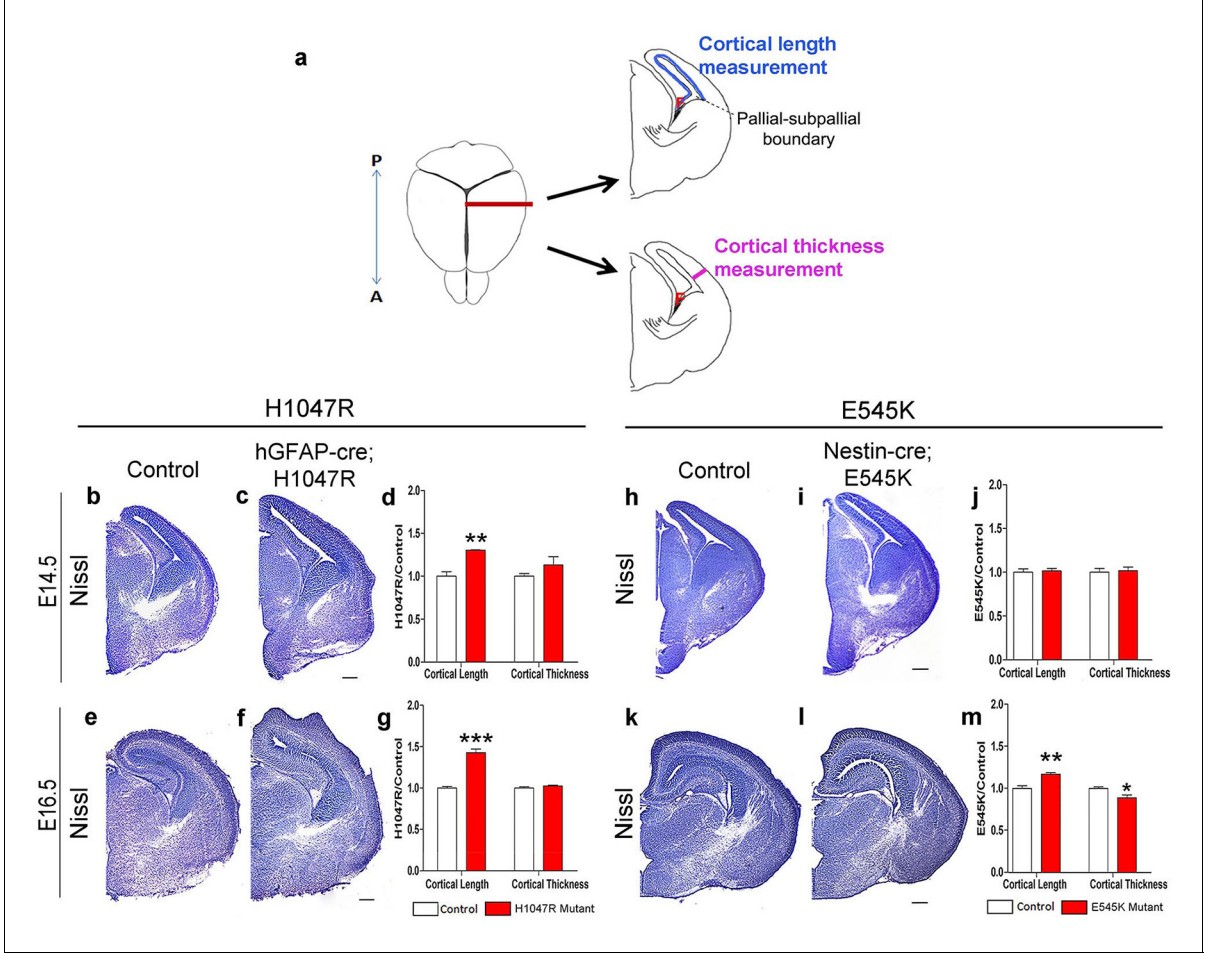

**Figure 2.** *Pik3ca* activating mutations lead to increased embryonic cortical length. (a) Schematic shows how cortical length and thickness were measured. F, fimbria/cortical hem. Nissl-stained coronal sections of control (b,e,h,k) and mutant (c,f,I,l) brains. (b–g) Cortical length of *hGFAP-cre; H1047R* mutant at E14.5 and E16.5 was longer than control; cortical thickness was not different. (h–m) Cortical length of *Nestin-cre;E545K* mutant was longer than control at E16.5 but not at E14.5; thickness was not different at E14.5 but was smaller than control at E16.5. Data are represented as mean ± SEM. *p<0.05; **p<0.01; ***p<0.001. Scale bars: 300 μm (b,c,e,f,h,i,k,l).

(YFP+) cells have the same size as controls (*Figure 3—figure supplement 2c,d*). We conclude that increased cell size due to *E545K* overactivation also has a critical embryonic period. Further, changes in multiple developmental parameters including proliferation, cell cycle exit, cell size and density contribute to MEG of *Nestin-cre;E545K Pik3ca* embryonic overactivation.

## Embryonic *Pik3ca* activation results in cortical dysplasia

Since disordered lamination is a key feature of human SEGCD (*Arai et al., 2012*; *Rossini et al., 2014*), we assessed neocortical organization and development in both *hGFAP-cre;H1047R* and *Nestin-cre;E545K* mutants. First, we studied the effect of *Pik3ca* overactivation on the Nestin-positive radial glial fibers, the scaffold for glial-guided neuronal migration, at multiple developmental stages. In *H1047R* mutants, the radial glial scaffold was slightly fasciculated and irregular at E14.5 and E16.5. Irregularities were very prominent at P0 when a disrupted pial surface was associated with irregular clusters of enlarged radial glial end-feet (*Figure 4—figure supplement 1*). The radial glial phenotype was much more subtle in the *E545K* mutant at E14.5 and E16.5; however at P0, we observed thinning of radial glial fibers and irregular clusters of end-feet at the intact pial surface (*Figure 5—figure supplement 1*).

Cajal-Retzius cells expressing Reelin, a major regulator of radial migration, were normally present in an ordered array in the marginal zone (layer I) of controls and *Nestin-cre;E545K* mice (*Figure 4b*;

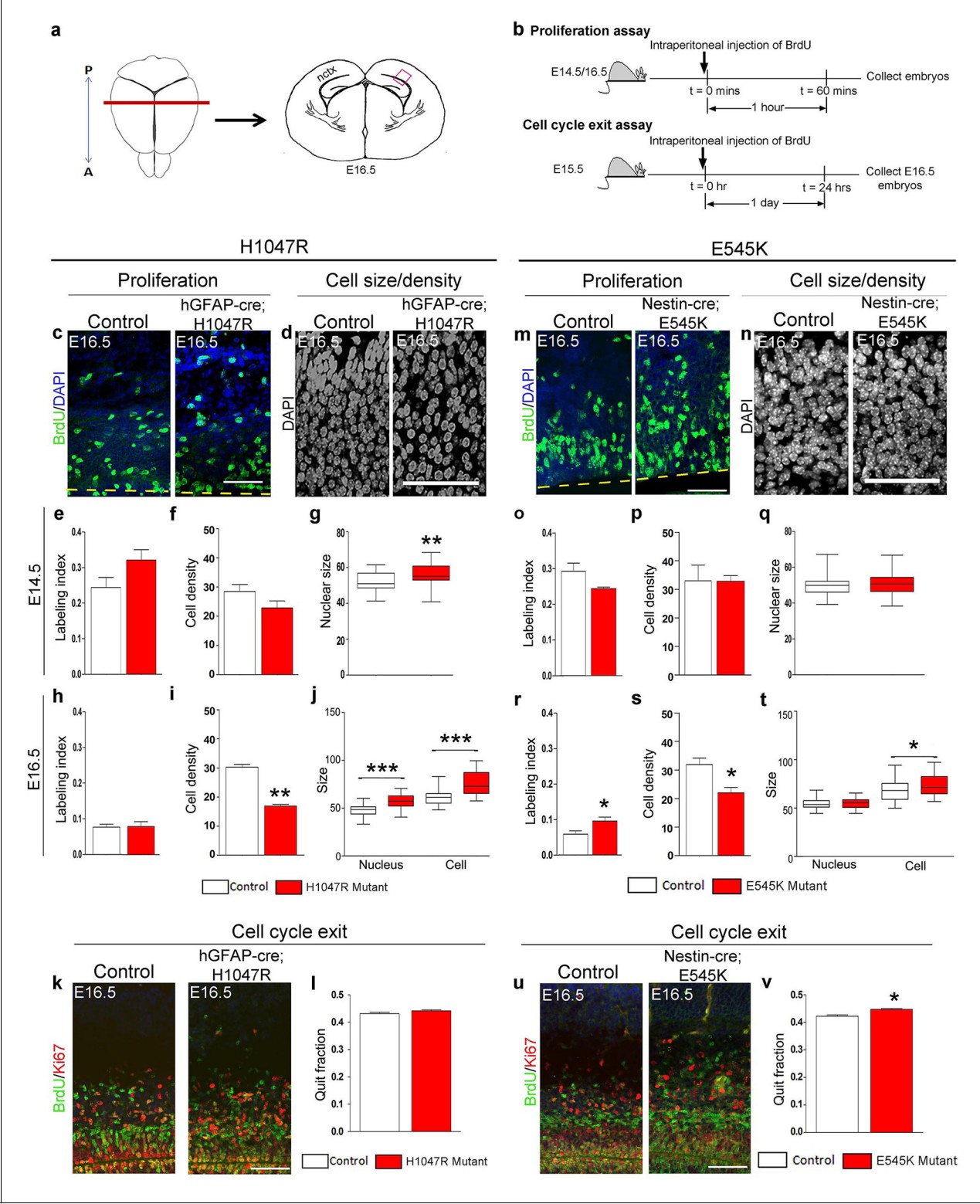

**Figure 3.** *H1047R* and *E545K* mutations differentially affect proliferation, cell density and size in neocortex. (**a**) Schematic shows area of interest (red box) in E16.5 mouse coronal section, as depicted in c,d,k,m,n,u. (**b**) Experimental outline of the proliferation and cell cycle exit assays. For labeling index, E14.5 and E16.5 control and mutant brains, harvested after a 1hr BrdU pulse, were processed for BrdU and DAPI staining (**c,m**). For quit fraction analysis, E16.5 control and mutant brains, pulsed with BrdU at E15.5, were processed for BrdU and Ki67 (**k,u**). Magnified view of DAPI-stained cortical nuclei shows differences in size and density between controls and mutants (**d,n**). (**e-j,l**) E14.5 and E16.5 *H1047R* mutants had similar labeling indices

*Figure 3 continued on next page*

Figure 3 continued

(BrdU[+] cells/total DAPI[+] cells); E16.5 *H1047R* mutant neocortex displayed reduced cell density (x10[5] DAPI[+] cells/mm[3] volume), larger nuclear and cell size ($\mu m^2$) and similar quit fraction (BrdU[+]Ki67[-] cells/total BrdU[+] cells). (o-q) E14.5 *E545K* mutant neocortex was similar to control in labeling index, cell density and nuclear size. (r-t,v) E16.5 *E545K* mutant showed significantly higher labeling index and quit fraction, reduced cell density, and enlarged cell and nuclear size, compared to controls. Data are represented as mean ± SEM (e,f,h,i,l,o,p,r,s,v) or as median-centered box-and-whisker plots (g,j,q,t); *p<0.05; **p<0.001; ***p<0.0001. Scale bars: 50 μm (c,d,m,n); 100 μm (k,u). See also *Figure 3—figure supplements 1–2*.
The following figure supplements are available for figure 3:

**Figure supplement 1.** Effect of *PIK3CA* mutations on total cell numbers and apoptosis.
**Figure supplement 2.** *E545K* mutation affects cell size when activated embryonically but not postnatally.

*Figure 5b,c*). However, these cells were dysplastic in *hGFAP-cre;H1047R* mice at E16.5 (*Figure 4c,c'*). We did not observe ectopic Reelin-positive cells within the cortical column in either mutant. As expected, within the developing wildtype neocortex, Ctip2 and Tbr1 were expressed predominantly in the early-born, deep layers (layers V-VI), while Cux1 was expressed in late-born upper layers (layers II-IV). *hGFAP-cre;H1047R* mutants displayed a marked disorganization of all layers. Ctip2/Tbr1-positive as well as Cux1-positive cells in the E16.5 *H1047R* mutant were dispersed throughout the cortical plate, with both early- and late-born neurons severely mislocalized (*Figure 4d–g*). Laminar disorganization was less severe in E16.5 *Nestin-cre;E545K* brains, but deep Ctip2/Tbr1-positive neurons and upper Cux1-positive neurons were dispersed throughout the cortical plate (*Figure 5f,g*).

Laminar patterns in postnatal animals remained disrupted in both mutants, with *hGFAP-cre;H1047R* cortex more affected *Nestin-cre;E545K* mutant cortex (*Figure 5h–m*). Thus a simple developmental delay was not the cause of dysplasia (*Figure 4h–m*; *Figure 5h–m*). In P3 *hGFAP-cre;H1047R* mutants, NeuN-positive mature cortical neurons were found within the normally cell-sparse marginal zone as well as in the cortical white matter and residual ventricular zone, a feature reported in SEGCD patients (*Arai et al., 2012*). Further, the cortical subplate was not readily discernible in these mutants (*Figure 4h–k*; *Figure 4—figure supplement 2e,f*), blurring the boundary between grey and white matter – a feature often observed in SEGCD patients (*Rossini et al., 2014*).

To determine whether the mislocalization of neocortical cells was due to defects in cell fate specification and/or migration, we labeled cells at either E12.5 or at E16.5 with pulse of BrdU and assessed cortical neuronal location and fate (layer V; Ctip2[+] and layers II/III; Cux1[+]) (*Figure 6a*). The total numbers of BrdU[+] P0 cells, born at E12.5 and E16.5, were not significantly different between controls and either *hGFAP-cre;H1047R* or *Nestin-cre;E545K* mutants (*Figure 6b,j*). The distribution of BrdU[+] cells showed significant differences between controls and *H1047R* mutants labelled during both early and late embryonic stages. At P0, more BrdU[+] cells were localized in the lower cortical plate (CP) and white matter (*Figure 6c*). For *E545K* mutants the BrdU[+] cell numbers were not different at either age. The distribution was subtly, yet significantly different only for the early born neurons (*Figure 6k*). For both the *H1047R* and *E545K Pik3ca* activating alleles, total layer V (Ctip2[+]) cell numbers at P0 were not significantly different between controls and mutants (*Figure 6d,l*). Also, the numbers of Ctip2/BrdU double-labeled cells were the same in controls and mutants, indicating that cell fate specification for these deep layer neurons was unaffected by either the *H1047R* or *E545K Pik3ca* allele (*Figure 6e,m*). However, similar to the overall BrdU[+] cell distribution, the specific distribution of layer V neurons was abnormal in *H1047R* mutants, with ectopic Ctip2/BrdU double-labeled cells in the upper and lower CP and white matter, instead of mid CP (*Figure 6f*). In *E545K* mutants, fewer Ctip2[+] cells were positioned in the mid CP (*Figure 6n*), although the phenotype was much less severe.

Upper layer (Cux1[+]) neuronal numbers and distributions were significantly different in both *H1047R* and *E545K* mutants, compared to their respective controls (*Figure 6g,o*). In *E545K* mutants, the increase in total Cux1[+] cell numbers in the *E545K* mutant corresponded to increased Cux1/BrdU double-labeled cells, born at E16.5 (*Figure 6p*). However, no such correlation was observed in E12.5 or E16.5-born Cux1[+] cells in the *H1047R* mutant (*Figure 6h*). These extra cells were therefore likely born between E16.5 and P0. The distribution of Cux1[+] cells was disrupted in both mutants, with the *H1047R* mutant displaying the more severe phenotype (*Figure 6i,q*). Together, these data indicate

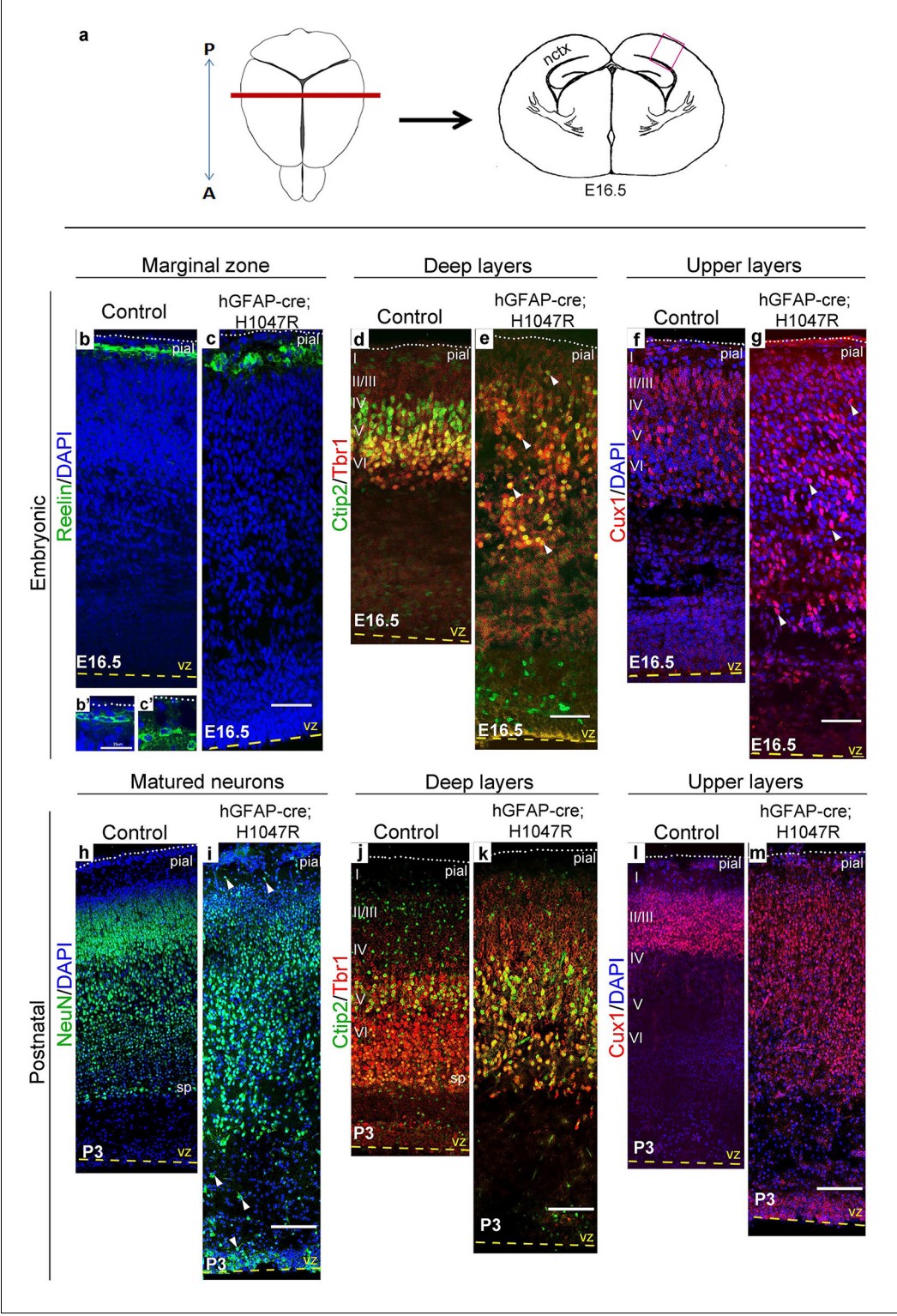

**Figure 4.** *H1047R* mutant mice display abnormal neocortical layering. (a) Schematics of mouse brain and section; section corresponds to the marked coronal plane; red box in the section marks the area of neocortex (nctx) depicted in the images below. (b–g) and (h–m) correspond to ages E16.5 and P3 respectively. In control cortex, Reelin is in layer I (b), Ctip2 and Tbr1 in deep layers VI and V (d,j), Cux1 in upper layers II-IV (f,l) and NeuN in all matured neurons (h). *H1047R* mutants displayed abnormal distribution of cells for all neocortical layers, observed at E16.5 and P3 (c,e,g,i,k,m). (b',c') Magnified view of Reelin-positive cells in control and *H1047R* mutant. P3 *H1047R* mutant showed enlarged area between ventricular zone (vz) and cortical plate and absence of clear subplate (sp) boundary (h-k). A, anterior; P, posterior; yellow dashed lines, lateral ventricular lining; white dotted

*Figure 4 continued on next page*

Figure 4 continued

lines, pial surface; I-VI, neocortical layers; arrowheads, mispositioned mutant cells. Scale bars: 25 µm (b',c'), 50 µm (b-i), 150 µm (j-o). See also *Figure 4—figure supplements 1–2*.

The following figure supplements are available for figure 4:

**Figure supplement 1.** Nestin expression in *hGFAP-cre; H1047R* mutant.

**Figure supplement 2.** *hGFAP-cre;H1047R* mutant displays distinct white matter dysplasia.

that cell fates are largely correctly specified in both *Pik3ca* mutants and that cortical dysplasia is more likely caused by aberrant neuronal migration.

## *Pik3ca* mutations cause white matter dysplasia

In P3 *hGFAP-cre;H1047R* mutants, although the cortical plate itself was not dramatically thicker than controls, the underlying cortical white matter was much thicker (*Figure 4j,k*; *Figure 4—figure supplement 2e,f*). This was less pronounced but readily discernible in P3 *Nestin-cre;E545K* mutants (*Figure 5—figure supplement 2c,d*). In P3 *H1047R mutants*, there was complete absence of corpus callosum, although hippocampal and anterior commissures were present (*Figure 4—figure supplement 1a-d*). In contrast, all major tracts were present in P3 *E545K* mutants (*Figure 5—figure supplement 2*). These data are consistent with the wide spectrum of white matter dysplasia reported in MEG and SEGCD patients (*Conway et al., 2007*; *Adamsbaum et al., 1998*; *De Rosa et al., 1992*; *Jansen et al., 2015*). Moreover, increased number of Olig2-positive cells was observed in the white matter area of both *H1047R* and *E545K* mutants (*Figure 4—figure supplement 2*, *Figure 5—figure supplement 2*). Although astrocytosis is observed when mTOR signaling is activated by TSC mutations in humans and mice (*Sosunov et al., 2008*; *Zeng et al., 2008*), it is not a feature of PIK3CA-pathology in our mouse models (*Figure 5—figure supplement 3a-d*).

## Both megalecephalic and normocephalic *E545K* mutant mice are epileptic

Epilepsy is one of the most important clinical features of SEGCD (*Bast et al., 2006*; *Fauser et al., 2015*; *Fauser, 2006*; *Arai et al., 2012*). Since most of the *H1047R* mutants were not viable post-weaning, we assessed *Nestin-cre;E545K* (megalencephalic) and *Nestin-creER;E545K* (normocephalic) adults for epilepsy phenotypes. Baseline sleep EEG recordings in both animal models revealed epileptiform activity including sets of spikes/polyspikes, and regional and generalized spike and wave discharges during non-rapid eye movement (NREM) sleep (*Figure 7b,c*). We also conducted additional 2 hr of continuous EEG recording immediately after 5 hr of total sleep deprivation of the *Nestin-creER;E545K* mice. Sleep deprivation is commonly implemented during epilepsy diagnostic studies in mice and humans and increases the sensitivity and specificity of EEG diagnosis for epilepsy (*De Rosa et al., 1992*; *Giorgi et al., 2013*; *Binnie and Prior, 1994*; *Kalume et al., 2015*). The frequency of epileptiform interictal activity was increased in post sleep deprivation EEG recordings, and clinically relevant spontaneous seizures including myoclonic (MC) seizures, frequent isolated spikes, and train of spikes, were observed in the *Nestin-creER;E545K* mice (*Figure 7d,e*).

When challenged with the chemoconvulsant pentylenetetrazol (PTZ), a GABA-A receptor antagonist (*Macdonald and Barker, 1978*), both the megalencephalic and normocephalic *E545K* mouse models exhibited lower seizure thresholds compared to controls at both P35 and P180 (*Figure 7f–i*; *Figure 7—figure supplement 1d*). In the 30 min post PTZ injection, both models showed shorter latencies to first generalized tonic clonic (GTC) seizures, more myoclonic seizures, and a prolonged seizure load.

We conclude that *Pik3ca* overactivation is sufficient to cause epilepsy. Further our data indicate that *Pik3ca*-related epilepsy is dissociable from brain overgrowth and cortical dysplasia.

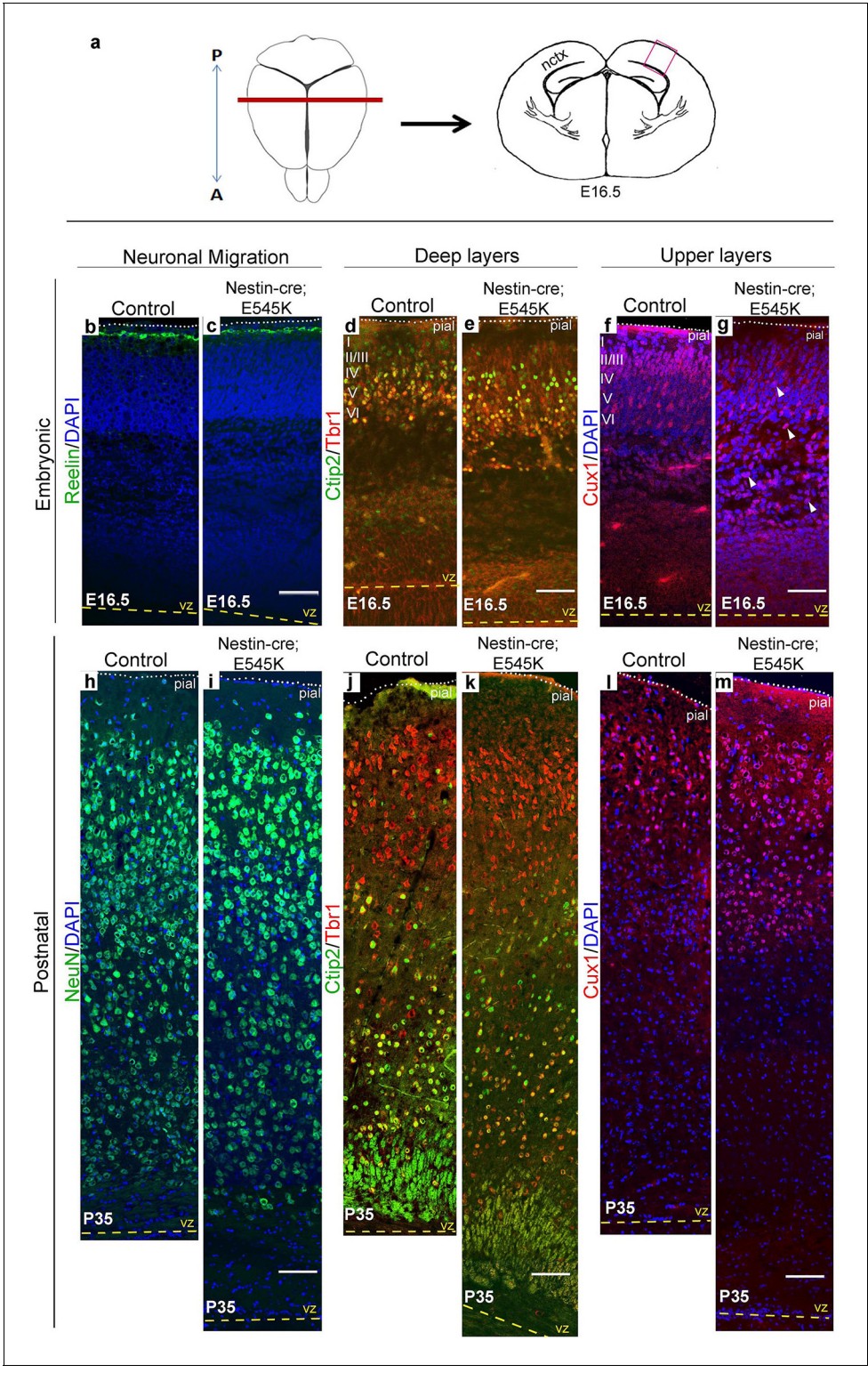

**Figure 5.** *E545K* mutant mice display abnormal neocortical upper layers. (a) Schematics of mouse brain and section; section corresponds to the marked coronal plane; red box marks the area of neocortex (nctx) depicted in the images below. (b-g) and (h-m) correspond to ages E16.5 and P35 respectively. (b-g) Compared to control, in E16.5 *E545K* mutant, layer I appeared normal; deep layers lacked normal arrangement while Cux1-positive cells were dispersed throughout the cortical plate. Extent of dispersion was reduced postnatally (h-m). vz, ventricular zone; yellow dashed lines, lateral ventricular lining; white dotted lines, pial surface; I-VI, neocortical layers; arrowheads, mispositioned mutant cells. Scale bars: 50 μm (b-g), 150 μm (h-m). See also *Figure 5—figure supplements 1–3*.

*Figure 5 continued on next page*

*Figure 5 continued*

The following figure supplements are available for figure 5:

**Figure supplement 1.** : Nestin expression in *Nestin-cre;E545K* mutant.

**Figure supplement 2.** *Nestin-cre;E545K* mutant displays distinct white matter dysplasia.

**Figure supplement 3.** Astrocytes show no gross dysmorphology in adult *Nestin-cre;E545K* and *Nestin-creER;E545K* mutants.

## Acute inhibition of Pik3ca activity suppresses epilepsy, rapidly altering cell signaling

BKM120, a 2,6-dimorpholino pyrimidine derivative, is an orally available pan-Class I PI3K inhibitor currently in clinical trials for solid tumors (*Maira et al., 2012*; *Bendell et al., 2012*; *Brachmann et al., 2012*) and may represent a novel therapeutic agent for *PIK3CA*-related epilepsy. Preclinical studies show that BKM120 maximally inhibits downstream phosphorylation of Akt, 1hr post-administration (*Maira et al., 2012*). To test its anti-seizure effects in our adult *Pik3ca*$^{E545K}$ gain-of-function megalencephalic and normocephalic models, we administered 50 mg/kg BKM120 (*Maira et al., 2012*) by oral gavage 1hr prior to PTZ-challenge at ~P35. BKM120 increased the seizure threshold of control animals. More importantly, despite the presence of megalencephaly and considerable cortical dysplasia in P35 *Nestin-cre;E545K* megencephalic animals, BKM120 dramatically decreased the seizure number and duration to untreated control levels and marginally increased seizure latency in the mutant mice (*Figure 7h,i*; *Figure 7—figure supplement 1d*). These data powerfully demonstrate that dynamic Pik3ca-dependent processes, independent of cortical and cellular dysplasia, cause *Pik3ca*-related epilepsy and they are highly amenable to therapeutic intervention.

To begin to dissect the cell signaling mechanisms underlying Pik3ca-driven epilepsy, we conducted reverse phase protein array (RPPA) analysis to measure protein levels of a comprehensive panel of cell signaling molecules (*Tibes et al., 2006*). We assessed subdissected cortical and hippocampal tissue from untreated (-) and PTZ, BKM120 and BKM120 PTZ treated adult control and *Nestin-cre;E545K* mutants (*Figure 8*, *Figure 8—figure supplement 1*). As expected, untreated *Nestin-cre;E545K* mutants exhibited significant elevations of phospho (p)S473-Akt and pT346-NDRG1, consistent with PI3K pathway activation. Notably, baseline pS473-Akt levels in the *Nestin-cre;E545K* hippocampus were prominently higher than the cortical levels. In both *E545K* mutant and control brain tissues, PTZ treatment alone increased the levels of pAkt, especially pS473-Akt, of pS6 (pS235/S236, pS240/244), pT346-NDRG1 and pS2448-mTOR. As expected, acute BKM120 treatment alone reduced phosphorylation of multiple PI3K pathway members, including AKT, S6, NDRG1, GSK3 and 4EBP1. Most remarkably, BKM120 also inhibited the increased phosphorylation levels induced by PTZ, notably returning mutant hippocampal pS473-AKT levels to baseline untreated control levels.

## Discussion

### *PIK3CA*-related disorders (PROS) in humans

Activating *PIK3CA* mutations have been associated with many human overgrowth disorders categorized based on severity and distribution of the mutation. Involvement of multiple tissues results in CLOVES or Klippel-Trenaunay syndrome with highly mosaic mutation levels (0.8–32%) in affected tissues (*Luks et al., 2015*; *Kurek et al., 2012*). Involvement of single tissue or body segment results in epidermal nevi, lymphatic malformations or other localized phenotypes with usually no mutations detected in unaffected tissues (*Keppler-Noreuil et al., 2014*; *Luks et al., 2015*; *Kurek et al., 2012*; *Osborn et al., 2015*; *Groesser et al., 2012*; *Lindhurst et al., 2012*; *Hafner et al., 2007*; *Rios et al., 2013*; *Cohen et al., 2014*). Too few patients and insufficient quantitative data have been reported to observe allele-specific differences.

In the brain, mosaic hotspot mutations result in SEGCD, classified as dysplastic MEG, HMEG or FCD2a based on extent of lesion (*Jansen et al., 2015*). *PIK3CA* mutations were detected in 9/73 patients with HMEG and 1/33 with FCD2 (*Lee et al., 2012*; *D'Gama et al., 2015*; *Jansen et al.,*

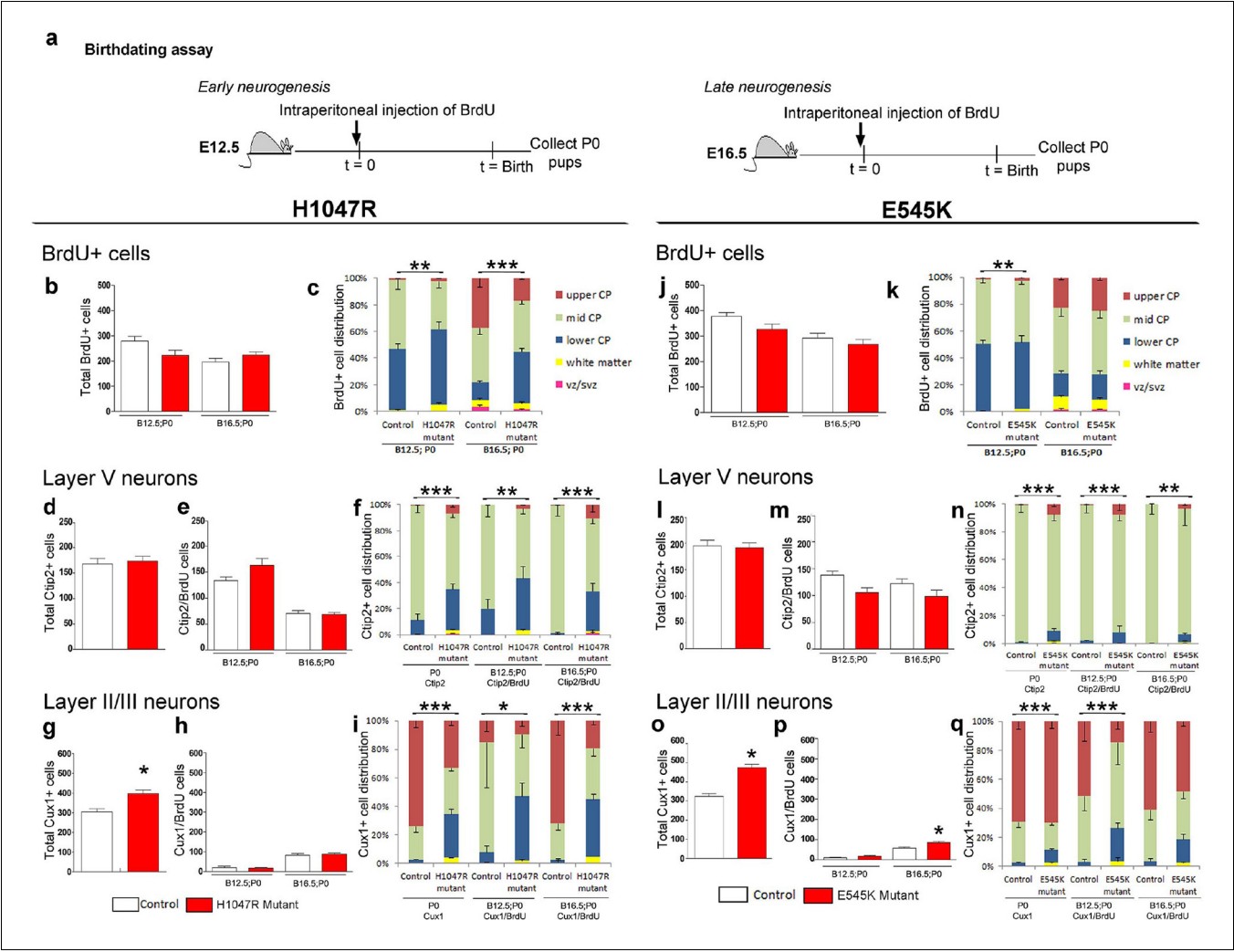

**Figure 6.** Birthdating assays demonstrate defects in laminar distribution. (a) Experimental outline of birthdating assays: BrdU was injected at E12.5 and E16.5 and analyzed at P0 (B12.5;P0 and B16.5;P0). Total number of BrdU+ cells at P0 generated at E12.5 and E16.5 (b,j), and total number of Ctip2+ cells (layer V neurons; d,l) were not significantly different between respective controls and mutants, for both *H1047R* and *E545K* lines. (c) Distribution of BrdU+ cells in the neocortex was significantly different between control and *hGFAP-cre;H1047R* mutant for both early and late assays, with more cells residing in the lower cortical plate and white matter instead of mid and upper zones of the cortical plate. (e,m) Total number of layer V neurons in both *H1047R* and *E545K* mutants, born at E12.5 and at E16.5, did not significantly differ from the respective controls; but showed significant difference in their zonal distribution with Ctip2+BrdU+ cells predominating the lower cortical plate in both the mutants (f,n). Total number of Cux1+ neurons (layers II/III neurons; g,o) was significantly higher in both the mutants compared with the respective controls. The colocalization of Cux1 and BrdU was not significantly different in the *H1047R* mutant and control for both ages (h); but number of Cux1+ cells born at E16.5 was significantly higher in *E545K* mutant than in the control (p). (i,q) Zonal distribution of Cux1+ cells was significantly different between controls and mutants, with more Cux1+ cells residing at the lower portion of the P0 cortical plate. The *H1047R* mutant phenotype is more extreme than the *E545K* mutant. Data are represented as mean ± SEM. *p<0.05; **p<0.001; ***p<0.0001.

*2015*). These overlapping SEGCD are associated with severe and usually intractable epilepsy (*Bast et al., 2006*; *Fauser et al., 2015*; *Fauser, 2006*; *Pasquier et al., 2002*; *Russo et al., 2003*; *Tassi, 2002*; *Devlin, 2003*; *Blümcke et al., 2011*). ~20 other *PIK3CA* mutant alleles have been seen in MCAP, characterized by MEG or MEG-PMG, hydrocephalus and less severe epilepsy (*Gymnopoulos et al., 2007*; *Mirzaa et al., 2012*; *Rivière et al., 2012*; *Tatton-Brown and Weksberg, 2013*).

We activated the two most common hotspot mutations, *E545K* and *H1047R*, in mouse brain at different developmental timepoints to generate the first models of human *PIK3CA*-related SEGCD.

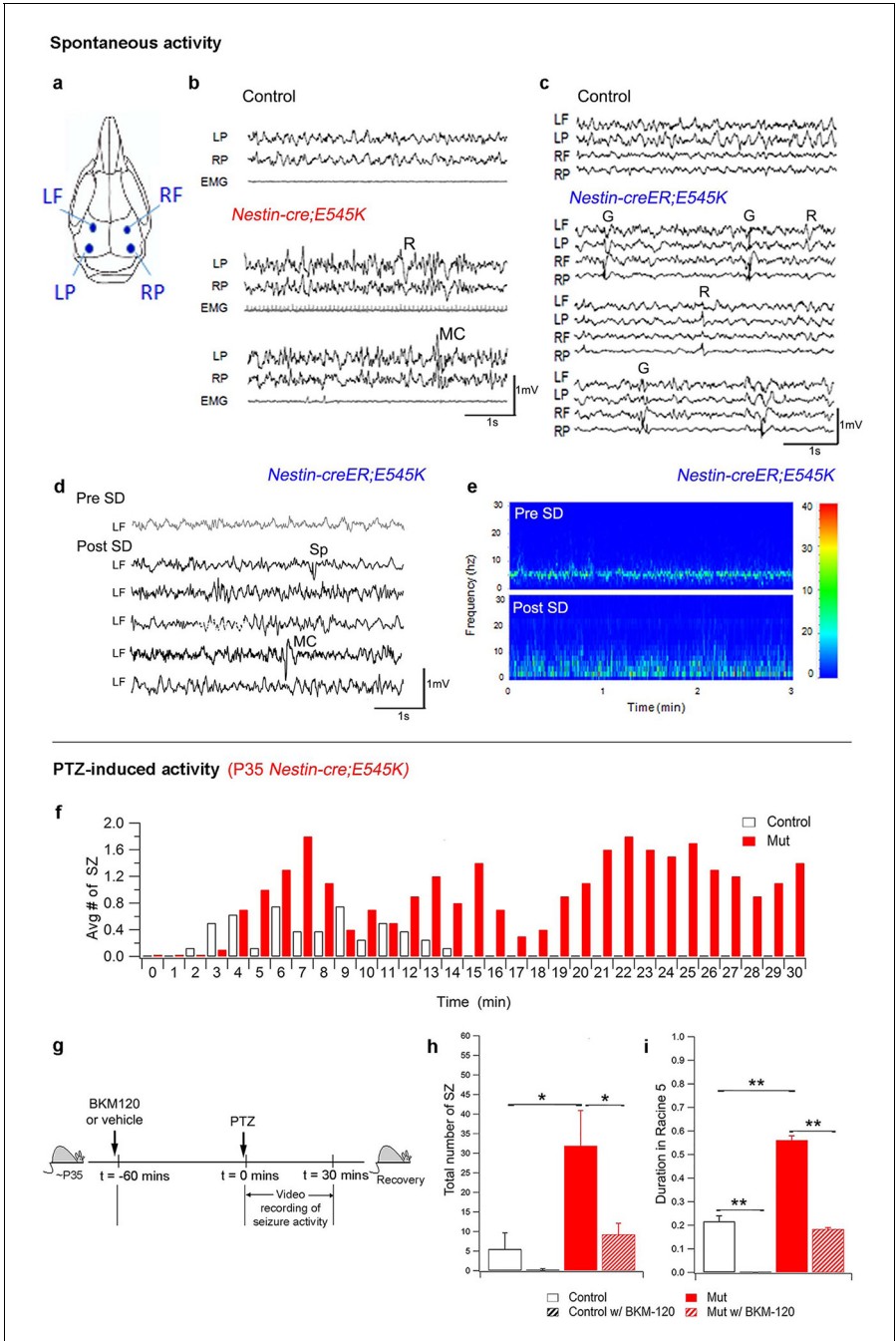

**Figure 7.** PI3K activity acutely modulates epileptic seizures. (**a**) Schematic shows electrode placement for EEG recordings. LF=Left Frontal, LP= Left Posterior, RF=Right Frontal, RP= Right Posterior. Only 2 electrodes were placed in P35 *Nestin-cre;E545K*. (**b**) EEG-EMG tracings of *Nestin-cre;E545K* mutant showed bilateral spikes/polyspikes, myoclonic (MC) seizures, fast and slow wave discharges, not associated with movement on video or EMG activity. (**c**) Generalized (G) and regional (R) spike and wave discharges were observed in *Nestin-creER;E545K* mice. Scale: 1s,1mV. (**d,e**) Sleep deprivation (SD) enhances epileptiform EEG activity in *Nestin-creER;E545K* mutant. EEG tracings of a *Nestin-creER;E545K* mutant mouse after 5 hr of normal sleep (Pre SD) and after 5 hr of total sleep deprivation (Post SD) in the same mouse (**d**), the mutant showing myoclonic (MC) seizures and isolated regional spikes (R). Power spectrum analysis, representing the frequency distribution for EEG activity over time, also displayed increased activity of the mutant post SD (**e**). (**f**) Bar chart showing average number of seizures (SZ) in PTZ-induced P35 *Nestin-cre;E545K* and control over time. (**g**) Experimental outline for BKM120-PTZ test. (**h**) Total number of seizures was significantly higher in P35 mutants than controls. Acute administration of BKM120 reduced number of seizures in mutants. (**i**) Duration of sustained generalized tonic-clonic seizure state (Racine 5), normalized to the total time of test, was significantly longer in P35 *Nestin-cre;E545K* mutants than controls. BKM120 significantly reduced the duration. Data are represented as mean ± SEM. *p<0.05; **p<0.0001. See also *Figure 7—figure supplement 1*.

*Figure 7 continued on next page*

*Figure 7 continued*

The following figure supplement is available for figure 7:

**Figure supplement 1.** Seizure activity of *E545K* mutants at old age.

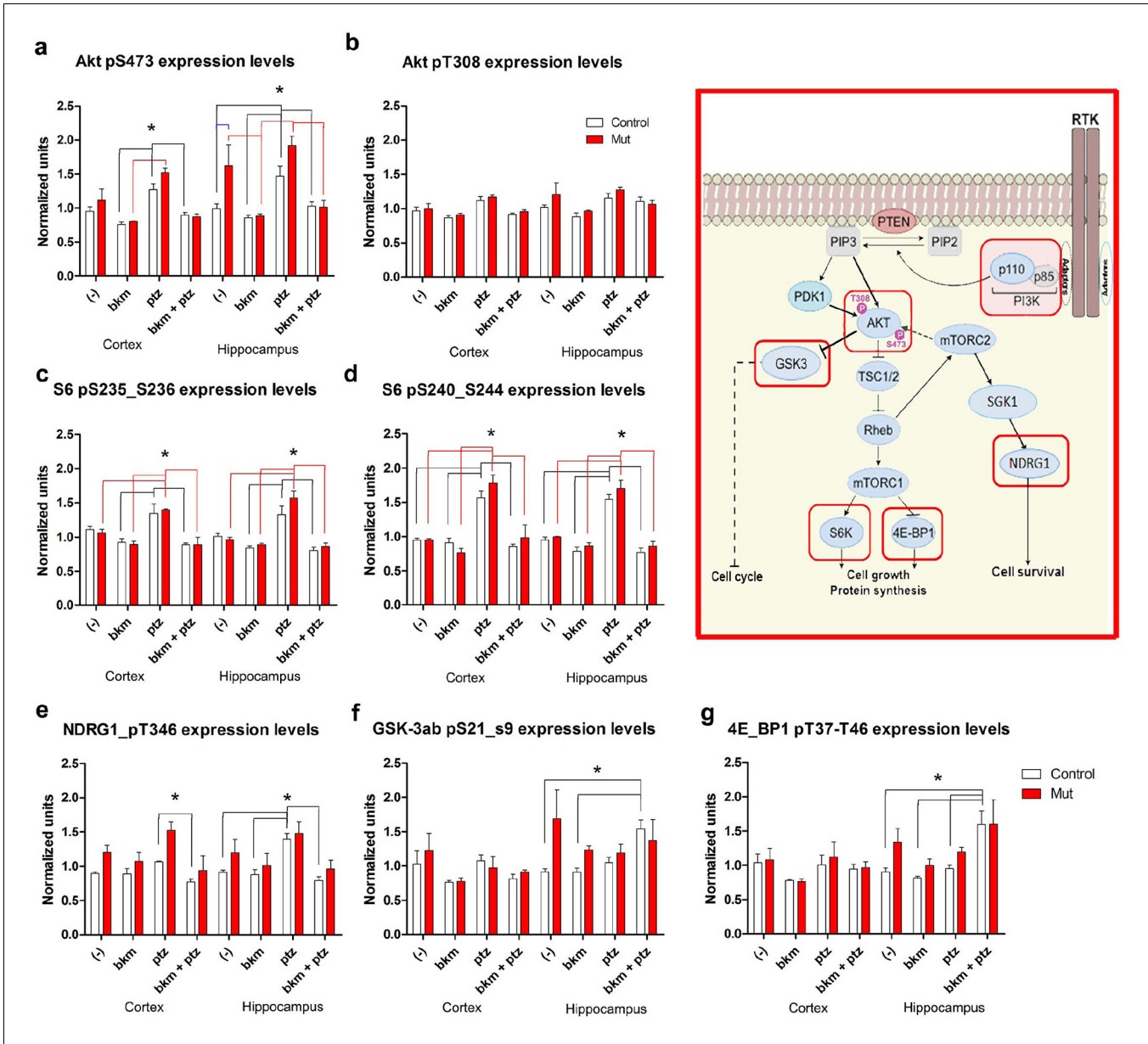

**Figure 8.** BKM120 acutely alter PI3K pathway protein profile. (**a–g**) Graphs show differential protein levels in P35 *Nestin-cre;E545K* mutant and control brains due to different treatments: untreated (-); BKM120; PTZ; BKM120+PTZ. Data are represented as mean ± SEM. *p<0.05. Inset shows simplified PI3K pathway; BKM120 significantly regulated the highlighted molecules. See also *Figure 8—figure supplement 1*.

The following figure supplement is available for figure 8:

**Figure supplement 1.** RPPA analysis graphs.

Our mouse models faithfully recapitulated the most important *PIK3CA*-related phenotypes of MEG, hydrocephalus, cortical and white matter dysplasia, and epilepsy (*Table 1*).

## Differential activating mechanisms underlie *Pik3ca* allele-specific brain phenotypes

Data from cancer biology suggests that *H1047R* mutation is more severe than *E545K* mutation. For example, *E545K* mutation accounts for 1932/7548 (26%) and *H1047R* for 2898/7548 (38%) of *PIK3CA*-coding mutations detected in the COSMIC database of cancer mutations (http://cancer.sanger.ac.uk/cancergenome/projects/cosmic).

We found that *H1047R* and *E545K* mutations caused distinct phenotypes in mice, *H1047R* being more severe than *E545K*. *hGFAP-cre;H1047R* mutants had severe hydrocephalus and died pre-weaning. In contrast all mice with the *E545K* allele survived through adulthood without hydrocephalus. Developmental analyses of *hGFAP-cre;H1047R* and *Nestin-cre;E545K* embryos revealed common mechanisms, such as larger neurons and lower cell densities, contributing to enlarged brain size, with differences more significant in *H1047R* mutants. *E545K* mutation also elevated cortical proliferation and cell cycle exit during late neurogenesis. We do not believe that overexpression of the transgenic *H1047R* allele versus the knock-in design of the *E545K* allele underlies the phenotypic differences. The PI3K enzyme is made of p110 (encoded by *Pik3ca*) and p85 subunits. p110 stability is entirely dependent on levels of p85 (*Geering et al., 2007*; *Fruman et al., 2000*; *Yu et al., 1998*) and we have not altered p85. Rather, the phenotypic differences more likely reflect distinct allele-specific overactivation of PI3K signaling. The *H1047R* mutation increases the level and duration of response to extracellular ligand, while *E545K* alters the helical domain resulting in constitutive low level signaling with a blunted response to extracellular ligands (*Miled et al., 2007*; *Zhao and Vogt, 2008*).

These differences likely reflect distinct mechanisms differentially altering PI3K signaling. Whereas *H1047R* mutation increases the level and duration of response to extracellular ligand, *E545K* alters the helical domain resulting in constitutive low level signaling with a blunted response to extracellular ligands (*Miled et al., 2007*; *Zhao and Vogt, 2008*).

## Effect of *Pik3ca* overactivation on brain and cell size is temporally regulated

By activating the *E545K* mutation in progressively limited progenitor pools, we decreased the size of brain and cells in a graduated fashion. Postnatal *E545K* activation had no impact on cell/brain size. We conclude that the *PIK3CA*-related brain overgrowth must arise from mosaic mutations in embryonic neural progenitors.

Although neuronal size was enlarged in both *hGFAP-cre;H1047R* and *Nestin-cre;E545K* mutants, it was less than that observed in *Pten null* mice or in cultured hippocampal neurons constitutively overexpressing *Akt* (*Kumar, 2005*; *Kwon et al., 2001*). Multiple models of *Pten* deletion cause

**Table 1.** Table displays comparison of the key features across different *Pik3ca* genetic models used in this study.

| Mutant allele | H1047R | | E545K | | |
|---|---|---|---|---|---|
| Cre driver | hGFAP-cre (dox from E0.5) | hGFAP-cre (dox from P1) | Nestin-cre | hGFAP-cre | Nestin-creERT2 (tamoxifen @P0/P1) |
| GoF expression onset | Late embryonic | Neonatal | Early embryonic | Late embryonic | Neonatal |
| Viability | Lethal by weaning age | Viable | Viable | Viable | Viable |
| Megalencephaly | ✓ | X | ✓ | ✓ (intermediate) | X |
| Hydrocephalus | ✓ | X | X | X | X |
| Increased cell size | ✓ | Not tested | ✓ | X | X |
| Cortical dysplasia | ✓ | X | ✓ | X | X |
| White matter dysplasia | ✓ | X | ✓ | ✓ (data not shown) | X |
| Epilepsy | Not tested | Not tested | ✓ | Not tested | ✓ |

progressive increases in postnatal neuronal size and increased brain size without continued proliferation (*Kwon et al., 2001*; *Fraser, 2004*; *Fraser et al., 2008*; *Backman et al., 2001*). Our *Nestin-cre;E545K* mutants had enlarged brain size evident at birth, without progressive increases in postnatal cell size. This is congruent with the analysis of resected human brain tissue from SEGCD patients. Mild cellular enlargement was observed with *PIK3CA* mutations in contrast to marked enlargement with *PTEN* or *AKT3* mutations (*Jansen et al., 2015*).

## Aberrant neuronal migration is a major contributor to Pik3ca-related cortical dysplasia

Brains of SEGCD patients show mild to moderate migration defects in early-born cortical neurons and more severe defects in late-born neurons (*Arai et al., 2012*; *Rossini et al., 2014*). Similarly, embryonic activation of *H1047R* and *E545K* in mice caused abnormal neocortical lamination, with late-migrating Cux1-positive neurons severely affected in both the mutants. Birthdating studies support the conclusion that Pik3ca activation does not alter cell fate and that cortical dysplasia is predominantly a result of aberrant migration. The severity of dyslamination in *H1047R* mutants likely reflects the dysplastic Reelin-positive Cajal-Retzius cells. However the Reelin-positive layer remained well defined in both mutants. This is in contrast to the ectopic Reelin expression in neurons expressing high levels of overactive pAKT introduced by electroporation into embryonic mouse cortex (*Baek et al., 2015*). In human SEGCD, late migrating neurons often fail to migrate to the upper layers (*Arai et al., 2012*), a phenotype more severe than seen in any of our mouse models. However, NeuN immunohistochemistry in *H1047R* mutants confirmed the presence of ectopic neurons in the subcortical white matter, as seen in human SEGCD brain (*Arai et al., 2012*; *Salamon, 2006*). Human MEG is associated with a wide range of white matter dysplasia ranging from agenesis of corpus callosum to thickening of subcortical axon bundles (*Conway et al., 2007*; *Adamsbaum et al., 1998*; *De Rosa et al., 1992*; *Jansen et al., 2015*; *Salamon, 2006*). These features were also faithfully recapitulated in our mouse models.

## *Pik3ca*-related epilepsy is an active Pik3ca-dependent process dissociable from dysmorphology

Both adult megalencephaic *Nestin-cre;E545K* and normocephalic *Nestin-creER;E545K* mice exhibited spontaneous seizures as well as lowered seizure thresholds upon PTZ-seizure induction. Although cortical dysplasia resulted from embryonic activation of *Pik3ca* in *Nestin-cre;E545K* mice, postnatal activation of *Nestin-creER;E545K* did not cause increased cell size or megalencephaly or altered cortical lamination. Thus Pik3ca-dependent epilepsy is independent of dysmorphology. Further, inhibitory interneurons were not grossly perturbed in *Nestin-creER;E545K* mice (data not shown). This is congruent with the fact that these interneurons are born at embryonic stages and their migration is almost complete before birth (*Batista-Brito and Fishell, 2009*). Therefore altered interneuron development in *Nestin-cre;E545K* may contribute to epilepsy, aberrant interneuron development cannot represent a common mechanism for epilepsy in both models. The observation that acute BKM120 treatment is sufficient to inhibit PTZ-induced seizures even in adult megalencephalic mice supports the argument that the epileptic seizures are independent of dysplasia since the latter is not reversed over the short course of treatment. This is an important finding since a large portion of FCD patients who do not show detectable dysplasia suffer from intractable epilepsy (*Bernasconi et al., 2011*).

Proteomic analyses of cell signaling networks in megalencephalic cortical and hippocampal tissue at baseline and treated with PTZ and/or BKM120 provide insight into the mechanism of Pik3ca-dependent epilepsy. *Nestin-cre;E545K* mutants had elevated PI3K signaling with a more robust upregulation of mTOR-dependent pS473-Akt than the direct PDK1-dependent pT308-Akt, similar to the recent findings in human brain samples with *PIK3CA* mutations (*Jansen et al., 2015*). The modest changes in signaling compared to controls is congruent with previous studies which demonstrated only modest changes in the steady-state levels of PI3K signaling in breast cancer cells with *PIK3CA* mutations (*Stemke-Hale et al., 2008*). Higher signaling levels in hippocampus versus cortex suggest a more prominent role of hippocampus in the seizure phenotype. PTZ administration alone in both controls and megalencephalic *Nestin-cre;E545K* mutants caused upregulation of many core components of PI3K-AKT pathway, including pAkt, pS6 and pNDRG1. This is congruent with a report

showing PTZ-induced seizures in rats upregulated PI3K-AKT-mTOR pathway (*Zhang and Wong, 2012*) and suggests that elevated baseline PI3K signaling levels are epileptogenic.

Indeed, there is extensive human and mouse evidence that elevated mTOR signaling is epileptogenic although the mechanisms for the epilepsy are incompletely understood. A number of mechanisms including altered development, cell size, growth, proliferation and circuitry have been reported (*Wong and Crino, 2012*). Most remarkably however, our acute BKM120 administration data clearly demonstrates that histopathological mechanisms are not the primary epilepsy drivers. Acute 1 hr of BKM120 administration was sufficient to completely inhibit the increased phosphorylation levels induced by PTZ, notably returning mutant hippocampal mTOR-dependent pS473-AKT levels to baseline untreated control levels. This was sufficient to normalize the PTZ-seizure induction threshold, despite continued dysplasia in *Nestin-cre;E545K* mutants. We conclude that elevated PI3K signaling is itself actively epileptogenic, independent of underlying developmental pathology.

### Changing the face of intractable pediatric epilepsy

The discovery that *Pik3ca*-related epilepsy is independent of dysplasia and susceptible to acute modulation is a major and paradigm shifting finding. Since PIK3CA resides at the top of the PI3K-AKT pathway, our mouse models represent surrogates for the entire group of patients with segmental brain overgrowth, including patients with somatic mosaic mTOR and AKT3 mutations (*Keppler-Noreuil et al., 2014*; *Lee et al., 2012*; *D'Gama et al., 2015*; *Conway et al., 2007*; *Jansen et al., 2015*; *Mirzaa et al., 2012*). SEGCD is associated with early onset, severe and frequently intractable epilepsy that responds poorly to standard seizure medications (*Bast et al., 2006*; *Fauser et al., 2015*; *Fauser, 2006*; *Pasquier et al., 2002*; *Russo et al., 2003*; *Tassi, 2002*). Epilepsy surgery has been comparatively more successful (73%) in combating seizures in the same children (*Fauser et al., 2015*). A drug-based therapy however, would clearly be preferable. mTOR inhibition with rapamycin has shown therapeutic promise in FCD patients and animals models (*Baek et al., 2015*; *Curatolo and Moavero, 2013*; *Lim et al., 2015*; *Moon et al., 2015*); however, rapamycin treatments are not acute. Our data demonstrates that acute small molecule-based modulation of PI3K signaling, despite the presence of dysplasia, has dramatic therapeutic benefit. This suggests that PI3K inhibitors offer a promising new avenue for effective antiepileptic therapy for large cohorts intractable pediatric epilepsy patients.

## Materials and methods

### Mice

The following mouse lines were used: *Nestin-cre* (Jackson Labs, Bar Harbor, Maine, USA; Stock #003771), *Nestin-creERT2* lines (Jackson Laboratory, Bar Harbor, Maine, USA, MGI:3641212 and line generated in SJB's lab, *Zhu et al., 2012*), human glial fibrillary acidic protein (*hGFAP*)-cre (Jackson Labs, Stock #004600), *Pik3ca^H1047R* transgenic (human *H1047R* transgene expression is under the control of a tetracycline-inducible promoter (TetO)) (*Liu et al., 2011*), *Rosa26-rtTA* line (Jackson Labs, Stock #005670), *Pik3ca^E545K* knock-in (*Robinson et al., 2012*), Ai-9 (Jackson Labs, Stock #007905), Ai-14 (Jackson Labs, Stock #007914), *R26-LSL-EYFP* (Jackson Labs, Stock #006148), *Rosa26-LacZ* (Jackson Labs, Stock #003474). We have designated the *Pik3ca^H1047R* and *Pik3ca^E545K* conditional mutant mice as *H1047R* and *E545K* mutants/lines throughout the manuscript.

All lines were maintained on a mixed genetic background, comprising of FVB, C57Bl6, 129 and CD1 strains. Noon of the day of vaginal plug was designated as embryonic day 0.5 (E0.5). The day of birth was designated as postnatal day 0 (P0). The *H1047R* and *Rosa26-rtTA* lines were intercrossed and female mice positive for both these alleles were crossed with *hGFAP-cre;RosartTA;Pik3ca^H1047R* males. To ensure that *cre* and *Pik3ca^H1047R* mutant transgene expression was correlated plugged females were treated with doxycycline (Sigma; 2 mg/ml) from E0.5 available *ad libitum* in drinking water. For the neonatal induction experiment, the pups were treated with doxycycline from P1. The *E545K* line was crossed to reporter lines to obtain *E545K* floxed allele and the reporter in the same mouse line. Tamoxifen (Sigma T5648) was dissolved at 37°C in corn oil (Sigma) at 5 mg/ml and was administered intraperitoneally to pups of the cross *Nestin-creER* X *Pik3ca^E545K* mice at a dose of 75 µg/g body weight, once a day at P0 and P1, to activate the *E545K* mutation postnatally. *hGFAP–cre*, *Nestin–cre* and *Nestin–creERT2* mice were genotyped by PCR using primers for the *cre*

coding region, as previously described (*Chizhikov, 2006*). Genotyping of other alleles were done according to the following references: *H1047R* and *Rosa+/-* (*Liu et al., 2011*), *E545Kfloxed/+* (*Robinson et al., 2012*), *EYFP/+* and *Ai9/+* (*Zhu et al., 2012*). All mouse procedures were approved by the Institutional Animal Care and Use Committees.

## Sample preparation and histochemical procedures

Embryos and postnatal pups were harvested in phosphate buffer saline (PBS); brains fixed in 4% paraformaldehyde (PFA) for 4 hr, equilibrated in 30% (wt/vol) sucrose made in PBS, and sectioned at 25 µm on a freezing microtome. Adult mice were perfused with 4% PFA, brains collected and fixed in 4% PFA overnight, sunk in 30% sucrose in PBS, embedded in optimum cutting temperature (OCT) compound and sectioned at 12 µm on a cryostat. Sections were then processed for Nissl, hematoxylin and eosin (H&E) or immunohistochemical staining.

*Immunohistochemistry*: Sections were washed thrice in PBS, boiled in 10 mM Sodium citrate solution for antigen retrieval, blocked in 5% serum in PBS with 0.1%Triton X-100 and then incubated overnight at 4°C with primary antibodies. The next day, sections were washed thrice in PBS, incubated with appropriate species-specific secondary antibodies conjugated with Alexa 488, 568, 594 or 647 fluorophores (Invitrogen) for 2 h at room temperature and then counterstained with DAPI to visualize nuclei. Sections were coverslipped using Fluorogel (EMS #17985) mounting medium. Immunostained sections were imaged in Zeiss LSM 710 Imager Z2 laser scanning confocal microscope using Zen 2009 software and later processed in ImageJ software (NIH, Bethesda, Maryland, USA). Primary antibodies used are: rat anti-BrdU (Abcam), mouse anti-BrdU (Roche), rabbit anti-Tbr1(EMD Millipore), mouse anti-Tbr2 (EMD Millipore), rat anti-Ctip2 (Abcam), rabbit anti-Cux1/CDP (Santa Cruz Biotechnology), rabbit anti-pS6 (Cell Signaling), mouse anti-NeuN (EMD Millipore), mouse anti-Reelin (EMD Millipore), rabbit anti-Laminin (Sigma), rabbit anti-Olig2 (EMD Millipore), mouse anti-Nestin (EMD Millipore), chicken anti-YFP (Abcam), rabbit anti-Ki67 (Vector Lab), mouse anti-S100 (Abcam).

*Nissl and H&E staining*: Sections were stained in 0.1% cresyl violet solution for 10 min, rinsed quickly in distilled water, dehydrated in 95% ethanol, and left in xylene before being coverslipped with Permount (Fischer Scientific). H&E staining was performed by passing the sections through Harris modified Hematoxylin solution (Fisher Scientific) and EosinY (Sigma) and then dehydrating them in increasing grades of ethanol before dipping in xylene and coverslipping. Brightfield images were taken in Leica MZFLIII microscope using Leica DFC425 camera and LAS V3.8 software.

## BrdU incorporation experiments

Bromodeoxyuridine (BrdU; Life Technologies) was administered intraperitoneally (100 µg/g of body weight) to pregnant mice at E14.5/16.5 for 1 hr, at E15.5 for 1 day and at E12.5/E16.5 for proliferation assays, cell cycle exit and birthdating experiments respectively. S-phase labeling index (LI) was calculated by dividing total BrdU$^+$ cells by total number of DAPI$^+$ cells. Quit fraction was calculated by dividing BrdU$^+$Ki67$^-$ cells by total number of BrdU$^+$ cells.

## β-Gal staining protocol

Brain sections were briefly fixed, washed in wash buffer at room temperature and then stained overnight at 37°C in the staining solution comprising of the X-gal substrate. The sections were then washed in wash buffer at room temperature and stored at 4°C.

## TUNEL staining

TUNEL staining was processed on E16.5 control and mutant sections using Roche In situ Cell Death Detection Kit, Fluorescein.

## Magnetic resonance imaging (MRI) for volumetric analysis

At least 5 mice of each genotype (age P40–60) were used for volumetric analyses. MRI study was performed using a 7 T Bruker ClinScan system (Bruker BioSpin MRI GmbH, Germany) equipped with 12S gradient coil. A 2-channel surface coil was used for MR imaging. Animals were anesthetized and maintained with 1.5% isoflurane during MRI sessions. Transverse T2-weighted turbo spin echo images were acquired for volume measurements (TR/TE = 3660/50 ms, FOV = 25 × 25 mm, matrix

= 320 × 320, NEX = 1, thickness = 0.4 mm, scan time = 6.5 min). Total brain volumes were obtained by manually segmenting brain regions from olfactory bulbs to cerebellum, and computing volumes using OsiriX (Pixmeo, Switzerland). Each data point in the graph represents 1 mouse.

## Seizure experiments

Mice obtained from the following crosses were used for experiments at ~P35 (young age) and ~P180 (old age): *Nestin-cre/+ X Pik3ca*$^{E545K}$*floxed/+* and *Nestin-creER/+ X Pik3ca*$^{E545K}$*floxed/+*. At least 5 animals of each genotype were used per treatment experiment.

*Pentylenetetrazole (PTZ) seizure test.* Mice were subcutaneously injected with PTZ (Sigma), a GABA (A) receptor-antagonist, at 40 mg/kg body weight and digital videos of the mice were recorded for 30 min post-PTZ injection. Principal behavior in each 10 second-bin of the recorded video was scored as 4 or 5 using the Racine scale of seizure severity (4, rearing with forelimb clonus; and 5, rearing and falling with forelimb clonus) (*Kalume, 2013*; *Racine, 1972*). *Treatment trials.* Pan-PI3K inhibitor BKM-120 (Novartis; 50 mg/kg body weight, dissolved in 0.5% Tween-80, 0.5% methyl-cellulose) or saline was administered by oral gavage to the mice 1hr before PTZ seizure test.

*Sleep deprivation (SD).* To permit control of circadian variations of sleep in these experiments, baseline (control) sleep data (Pre SD) were recorded from mice one day before they were submitted to total sleep deprivation. Mice were allowed to sleep normally for 5 continuous hours beginning at 8:00 AM, and then baseline sleep video-EEG recordings were obtained continuously in the 1 subsequent hour. On the following day, beginning at 8:00 AM, the same mice were kept awake for 5 consecutive hours by random gentle touches with a rotating light curtain attached to a motor mounted on the lid of the sleep deprivation chamber. The motor was in turn, connected to a computer via Power Lab (ADInstruments, Colorado Spring, CO). The random direction and speed of the motor rotation were custom-programmed in the stimulator panel dialog box of LabChart 8 Software (ADInstruments, Colorado Spring, CO). The specific parameters used are tabulated as *Supplementary file 1*. Post sleep deprivation, mice were not disturbed and post SD recordings were obtained for 2hr.

*Video-electroencephalagraphy-electromyography (Video-EEG-EMG) recording.* These experiments were performed as previously described (*Kalume, 2013*). Briefly, mice underwent survival surgery to implant fine (diameter: 130 µm bare; 180 µm coated) silver wire EEG and EMG electrodes under isoflurane anesthesia. Four EEG electrodes were placed bilaterally through the small cranial burr holes over the posterior and frontal cortices and were fixed in place with cyanoacrylate glue and dental cement (Lang Dental Manufacturing Co., Inc., Wheeling, IL). Similarly, one reference electrode was placed above the cerebellum. A ground electrode was inserted subcutaneously over the back. EMG electrodes were placed in back muscles. Only 2 electrodes were implanted in the young *Nestin-cre;E545K* mutant and control mice. Mice were allowed to recover from surgery for 2-3 days. Simultaneous video-EEG-EMG recordings were collected from conscious mice on a PowerLab 8/35 data acquisition unit using LabChart 7.3.3 software (AD Instruments, Colorado Spring, Co). All bioelectrical signals were acquired at 1KHz sampling rate. The EEG signals were processed off-line with a 1-70 Hz bandpass filter and the ECG signals with a 3-Hz highpass filter. Interictal spikes were identified as transient, clearly distinguished from background activity, with pointed peak and short duration. Myoclonic seizures were identified as shock-like jerks of the muscles on video associated with a spike or polyspike-wave complex on EEG.

## Reverse phase protein array (RPPA) analysis

Cortex and hippocampus were dissected out of P35 control and *E545K* mutant mice, following different treatments (vehicle only (-), +BKM120, +PTZ, BKM120+PTZ), and flash-frozen in liquid nitrogen then sent to the RPPA Core Facility at MD Anderson Cancer Center, University of Texas. Three independent biological replicates per sample were analyzed. Analysis was performed as previously described (*Tibes et al., 2006*). The mouse brain tissue samples were lysed and underwent protein extraction. Cellular protein was denatured by SDS sample buffer and serial dilution was made for each sample. Cell lysates were then probed with different validated antibodies. Signals were detected by DAB colorimetric reaction and intensity was quantified using ArrayPro software. Protein concentration was determined by super curve fitting. All the data points were normalized for protein loading and transformed to linear value. These linear values were used to make bar graphs for comparative analysis. See *http://www.mdanderson.org/education-and-research/resources-for-*

*professionals/scientific-resources/core-facilities-and-services/functional-proteomics-rppa-core/index. html)* for a detailed antibody list and protocols.

## Quantitative analysis

For quantitative analysis of embryos, data was collected from comparable sections of a minimum of 3 embryos of each genotype (from 2 or more independent litters) at each developmental stage. Cortical length was measured in the lateral ventricular lining from the tip of the fimbria/cortical hem to the pallial-subpallial boundary. Cortical length and thickness were measured using ImageJ software (NIH, Bethesda, Maryland, USA); the data was normalized to the control value. Cell counts from E14.5 and E16.5 brains were obtained from 25% of the neocortex. Area of interest was derived by dividing the whole length of neocortex into quarters and then taking images of the total area, from pia to ventricle, in the third quartile from dorsal midline. Confocal stacks of immunostained sections of each developmental stage were generated by scanning at intervals of 0.99 µm using filters of appropriate wavelengths at 20X and 40X magnifications. Confocal images of DAPI-stained brain sections and NeuN/pS6-immunostained sections were used to measure nuclear and cell size respectively. Measurements for labeling index, quit fraction, birthdating studies, cell density and size were calculated using ImageJ. For zonal quantification of cells, the cortical column was divided into 5 different parts – the ventricular-subventricular zone (vz/svz), white matter, and 3 equally divided zones of the neocortical plate (lower, mid, upper).

Statistical significance was assessed using 2-tailed unpaired t-tests (for cortical length and thickness, cell density, nuclear size, TUNEL assay, labeling index, quit fraction, total cell counts and seizure data) and ANOVA followed by Bonferroni (for cell size, BKM treatment data, birthdating experiments) and Tukey (for RPPA graphs) post-tests. These analyses were performed in GraphPad Prism v5.01 (GraphPad Software Inc., San Diego, USA) or in Igor Pro v6.3.6.4, Igor Pro Software, Lake Oswego, USA. Differences were considered significant at $P < 0.05$.

## Acknowledgements

We thank Leon Murphy (Novartis) for BKM120, Taylor Faubion for technical support, Joseph Gleeson for sharing unpublished data and Ghayda Mirzaa and Laura Jansen for discussions.

## Additional information

### Funding

| Funder | Grant reference number | Author |
|---|---|---|
| Seattle Children's Hydrocephalus Research Guild | Seattle Children's Hydrocephalus Guild Seed Funds | Kathleen J Millen |
| National Institutes of Health | R01:NS072441 | Kathleen J Millen |
| National Cancer Institute | CCSG:CA16672 | Gordon B Mills |
| Citizens United for Research in Epilepsy | SUDEP grant 2014, Christopher Donalty and Kyle Coggins Award | Franck Kalume |
| National Institutes of Health | R01:NS058721 | William B Dobyns |
| National Institutes of Health | R01:CA188516 | Suzanne J Baker |
| National Institutes of Health | P50:CA015962-02 | Jean J Zhao |
| National Institutes of Health | P01:CA142536 | Jean J Zhao |

The funders had no role in study design, data collection and interpretation, or the decision to submit the work for publication.

### Author contributions

AR, JS, FK, KJM, Conception and design, Acquisition of data, Analysis and interpretation of data, Drafting or revising the article; JN, Conception and design, Acquisition of data, Analysis and

interpretation of data, Contributed unpublished essential data or reagents; SR, Conception and design, Acquisition of data, Analysis and interpretation of data; YL, Acquisition of data, Analysis and interpretation of data; WBD, GBM, Conception and design, Analysis and interpretation of data, Drafting or revising the article; JJZ, SJB, Conception and design, Analysis and interpretation of data, Contributed unpublished essential data or reagents

## Ethics

Animal experimentation: All animal experimentation done in this study was done in accordance with the guidelines laid down by the Institutional Animal Care and Use Committees (IACUC) of Seattle Children's Research Institute, Seattle, WA (protocols 14208 and 14395), St. Jude Children's Research Hospital, Memphis, TN (protocol 278), and Dana Farber Cancer Institute, Boston, MA (protocol 06-034).

## Additional files

### Supplementary files

• Supplementary file 1. Parameters for motor rotation in LabChart 8 Software Table shows the list of parameters customized in the stimulator panel dialog box of LabChart 8 Software, in order to randomize the speed and direction of rotation of the motor used for the sleep deprivation study. The stimulation program cycle, comprising segments 1–12, was repeated through the entire duration of the sleep deprivation experiment (i.e. for 5 hr).

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
