## [Decision Letter]

Thank you for choosing to send your work entitled "Mouse models of human PIK3CA-related segmental overgrowth have treatment-responsive epilepsy" for consideration at *eLife*. Your full submission has been evaluated by Sean Morrison (Senior Editor) and three peer reviewers, one of whom is a member of our Board of Reviewing Editors, and the decision was reached after discussions between the reviewers. Based on our discussions and the individual reviews below, we regret to inform you that your work will not be considered further for publication in *eLife*.

In particular, while all three reviewers agreed that the work was potentially of interest, they felt that the characterization of the developmental phenotype was insufficient, and that it would require extensive new experimentation to make it rigorous enough for publication. In addition, the reviewers were concerned about the lack of insight into the epilepsy phenotype, with regard to cell biological mechanisms.

*Reviewer #1*:

In this manuscript by Roy et al., the authors have examined the effects of hyperactivating PI3-kinase mutations on neural development, have compared these effects to those seen in humans carrying similar mutations, and have shown that inhibition of PI3-kinase activity can rescue the seizure phenotype seen in adult mice. In general, this is a nicely-performed study and the conclusions are interesting from a variety of perspectives. However, this manuscript could be significantly improved by a consideration of the following issues.

1) The authors have used a number of Cre driver lines to look at two different activating mutations in PIK3CA, H1047R and E545K. However, the strategies for driving expression of these mutant alleles are very different, with the former involving tet-inducible gene expression and the latter a cre-dependent knock-in allele. Since the *E545K* strategy mimics the human congenital mutations that the authors seek to model, and the *H1047R* is essentially an overexpression study and thus not as relevant, then the authors should really emphasize (and show) all of their findings with the *E545K* mutant, even though in many cases the differences are not as dramatic. For example, the experiments documenting neuronal colocalization with *E545K* are all shown in the supplementary data while those with the *H1047R* mutant are shown in the body of the paper in Figure 4. This should be reversed.

2) Some of the most interesting comparisons in this manuscript are those looking at different timepoints when the activating allele is expressed. However, in order to validate these comparisons, the authors need to show a comparison of the level of recombination that they are driving. I know that these are well-characterized cre driver lines, but having said that, there has been some drift with a number of these lines (for example the *Nestin-cre* mouse used here), and the authors really should provide their own data to validate the mutants they are studying.

3) The section on neuronal mislocalization is very interesting, but needs to be strengthened in a number of additional ways. First, the authors need to provide some kind of quantification, focusing on the *E545K* mutants, to solidify their data. This could be as simple as dividing the cortex into zones/layers (depending upon the age), and quantifying the proportion of layer-specific marker-positive cells that are in the different zones/layers in controls versus mutants. Second, the authors should remove their references to neuronal migration. They don't actually study migration anywhere in this manuscript, but instead study localization of neurons. Since, as they indicate in the discussion, these phenotypes could be due to many different developmental perturbations, to indicate that they are due to neuronal migration deficits is misleading.

4) The approach of RPPA to scan a large number of signaling proteins at once is a valuable one. Nonetheless, it is important that these changes be validated using other approaches. In particular, the authors should choose at least a few of the more important changes and verify them using western blots. If they all match the data obtained using RPPA, then this will validate the entire approach for these particular samples.

*Reviewer #2*:

In this study the authors create animal models of megalencephaly (MEG) that are associated with mutations in PIK3CA. They first show that overexpression of PIK3CA in neural progenitors disrupts normal cortical development, and demonstrate that the timing of transgene activation is a critical feature of disease severity. They also demonstrate that overexpression of the two PIK3CA alleles leads to increased size of the cortex that is generally associated with an increase in nuclear and cell size and a decrease in cell density. They then demonstrate that the expanded cortex displays defects in lamination. As a consequence, transgenic animals overexpressing active alleles of PIK3CA are more prone to epileptic seizures. Finally, the authors show that a class I PI3K inhibitor has anti-seizure effects, and reduces phosphorylation of PI3K pathway components.

1) Validation of animal models. The authors need to show exactly when and where the p110a transgenes are expressed by their *Nestin-cre, Nestin-creER* and *hGFAP-cre* driver lines. Could there be differences in phenotypes from these different cre drivers based on level of expression, rather than simply due to the timing of cre activation? Also, the timing of tamoxifen administration for nestin-creER should be given in the results section to know when the transgenes were induced with this driver.

2) Comparisons of animal models. The authors do a nice job of presenting the different phenotypes associated with overexpression of the *E545K* allele achieved with the different *cre* drivers, but it would help to have more information on the *H1047R* allele. How does overexpression of *H1047R* affect brain size and morphology with the *Nestin-creER* and *Nestin-cre* lines? The authors stated that the most severe phenotype was achieved in *hGFAP-cre;H1047R* mtuants, but relative to what? To the other *H1047R* lines, or relative to *E545K* lines (or both)? If *H1047R* was only tested with the *hGFAP-cre* line, this should be clearly stated.

*Reviewer #3*:

This manuscript describes the phenotype of mouse models with mutations in PIK3CA that should overactivate the PI3K-AKT pathway in the nervous system. The *E545K* and *H1047R* activating mutations have been found before in patients with cancer, but also with various forms of segmental cortical dysplasia. Corresponding mouse lines have been generated and investigated before for studying the role of these activating mutations in cancer. In this study, the authors have used *Nestin-Cre* and *GFAP-Cre* for nervous system-specific induction of the mutant alleles, and these mice developed enlarged brain size, cortical dysplasia and epilepsy. The authors find that *GFAP-Cre H1047R* mutants show a very severe phenotype with progressive hydrocephalus and death around birth. Mice with *E545K* mutation and the same *GFAP-Cre* driver were less affected. Earlier developmental induction of the *E545K* mutation with *Nestin-Cre* showed a more severe phenotype than that observed after *GFAP-Cre*-mediated activation. The authors then found that the inner cortical length of *GFAP-Cre H1047R* mutants at E14-E16 was longer than in controls. Surprisingly, increased proliferation was not observed in these animals, only enlarged cell size. Interestingly, the *Nestin-Cre E545K* mutants showed enhanced proliferation with reduced cortical thickness, indicating that the newborn cells did not adequately migrate. The authors also observed dysplastic Cajal-Retzius cells in *GFAP-Cre H1047R* mice at E16. They then investigated *Nestin-Cre* and Tamoxifen-inducible *Nestin-Cre E545K* mutant mice for abnormalities in EEG and found signs of epileptiform activity, in particular when induced with the GABA-A receptor antagonist PTZ. The authors then also provide evidence that the PI3K inhibitor BKM-120 suppresses seizure activity and shows some effects on parameters of PI3K pathway activation.

Although these data are of potential interest to understand the molecular basis of segmental cortical dysplasia in humans, the paper is largely descriptive and the mechanisms how PI3K activation induces this complex disorganization of cortical development that then results in altered excitability and epileptic seizures is not explained. It is even unclear whether alterations in cell division, cell cycle exit, migration, terminal differentiation, spontaneous activity and synaptic plasticity occur in these mice. It is unclear whether *GFAP-Cre* and *Nestin-Cre* act in the same or in distinct precursor cell populations, whether defects in all types of neurons, or only specific subtypes, or defects in astrocytes cause this specific phenotype. The latter is possible in particular in *GFAP-Cre* mice. The previous paper by Robinson et al. in Nature 2012 showed that the same E545K mutation, when induced with *Blbp-Cre* in lower rhombic lip progenitors accelerates tumor generation, but does not per se initiate WNT subgroup medulloblastoma. This leaves the question open whether the same mutation in *Nestin*- or *GFAP-Cre* expressing cells causes defects that need additional signals to develop the pathogenic phenotype.

Additional specific comments:

1) The paper lacks clear evidence about the types of cells in which the mutant PIK3CA isoforms are expressed, and at which stage of development recombination occurs. This is necessary to explain whether dysregulated PIK3CA causes abnormalities in division of neural precursor cells. I wonder why the authors did not do more detailed pulse chase experiments with BrdU to study potential effects on altered proliferation of different types of precursor cells in the ventricular and subventricular zone. It would also be important to know whether proliferation of neuronal cells is differentially affected in comparison to proliferation of astrocytic precursor cells. At E16.5, the time point when the analyses shown in Figure 2 were performed, proliferation of astrocytes has already started, and the authors should do the studies shown in Figure 3 with markers that can distinguish neuronal from astrocytic precursor cells.

2) It is hard to explain why the cell density, as shown in Figure 3 is reduced when BrdU incorperation is increased. It is also not clear in which layers of the cortex the authors observe this reduced cell density at E16.5. To understand this phenotype, the VZ/SVZ, the mantle zone and the intermediate zone should be investigated separately. Did the authors also investigate whether there is enhanced loss of cells by apoptosis or other types of cell death between E14.5 and E16.5 to explain this phenotype?

3) The phenotype with increased cell size is hard to explain. This could be due to a defect in the cell fate determination of distinct populations of neuronal types, for example when generation and survival of pyramidal cells in comparison to interneurons is enhanced, or also by altered generations of neurons in comparison to glial cells. In order to understand the phenotype of altered cortical morphology and enhanced epileptic seizures, this would be important to know.

4) There is a big gap between the defects in cortical thickness and the potential mechanisms that could explain the altered threshold for epileptic seizures. Is there a dysbalance between GABAergic and glutamatergic neurons and synapses in the brain of these mice? Is there defective regulation of synapse generation, function and plasticity? Are there defects in dendritic arborization and synaptogenesis? Are there differences in the morphology of astrocytes and differences in gene expression in such astrocytes, i.e. for neurotransmitter transporters, that could explain the epileptiform activity?

5) It is not clear whether the enhanced epileptiform activity reflects a developmental defect or altered synaptic activity, which is directly driven by altered PIK3CA activation. The authors have used both a constitutive *Nestin-Cre* and a Tamoxifen-inducible Cre driver, but there are no data to distinguish the phenotype when E545K mutant PIK3CA is expressed from developmental stages, or only after acute induction in the adult.

Other questions and points of concern:

1) In the Materials and methods section, the authors say that all mouse lines were maintained on a mixed genetic background. It is not clear what this means and whether differences in genetic background could influence the phenotype. How can the authors exclude the possibility that the more severe phenotype observed in the *GFAP-Cre* drivers in comparison to *Nestin-Cre* drivers is due to different genetic background?

2) The data shown in Figure 5 are very hard to understand and not well described. What is the function of time at which the numbers of seizures occur? Does the diagram mean seizures within one minute? These bars lack standard deviations. How can one understand i.e. 0.4 seizures or 1.6 seizures per minute and why does the number of seizures fluctuate with a peak at 7 min., 15 and 22 min. after PTZ injection? Which phase of seizure activity is affected by BKM-120-treatment? Obviously, the PI3K inhibitor does not reduce seizure activity completely.

3) The data shown in Figure 6 are hard to understand, given that the differences are generally much lower than expected, and it is also not clear which types of cells and which subpopulations of neurons are predominantly affected by altered Akt phosphorylation or other downstream signaling parameters that are expected to be altered when the PI3-K pathway is constitutively activated.

---

## [Author Response]

Reviewer #1:

*In this manuscript by Roy et al., the authors have examined the effects of hyperactivating PI3-kinase mutations on neural development, have compared these effects to those seen in humans carrying similar mutations, and have shown that inhibition of PI3-kinase activity can rescue the seizure phenotype seen in adult mice. In general, this is a nicely-performed study and the conclusions are interesting from a variety of perspectives. However, this manuscript could be significantly improved by a consideration of the following issues. 1) The authors have used a number of Cre driver lines to look at two different activating mutations in PIK3CA, H1047R and E545K. However, the strategies for driving expression of these mutant alleles are very different, with the former involving tet-inducible gene expression and the latter a cre-dependent knock-in allele. Since the E545K strategy mimics the human congenital mutations that the authors seek to model, and the H1047R is essentially an overexpression study and thus not as relevant, […]*

The *H1047R* allele is *not* functionally an overexpression allele. The PI3K enzyme is made of p110 (encoded by Pik3ca) and p85 subunits. Previous publications have demonstrated that p110 stability is dependent on levels of p85 (Fruman et al., 2000; Yu et al., 1998). Geering et al., 2007 demonstrated that there is no evidence of free p110 or free p85 subunits in mammalian cell lines and tissues. Thus, both the *E545K* and *H1047R* alleles are relevant to model human *PIK3CA* mutations.

*[…] then the authors should really emphasize (and show) all of their findings with the E545K mutant, even though in many cases the differences are not as dramatic. For example, the experiments documenting neuronal colocalization with E545K are all shown in the supplementary data while those with the H1047R mutant are shown in the body of the paper in Figure 4. This should be reversed.*

We have included data from both alleles throughout the main figures.

*2) Some of the most interesting comparisons in this manuscript are those looking at different timepoints when the activating allele is expressed. However, in order to validate these comparisons, the authors need to show a comparison of the level of recombination that they are driving. I know that these are well-characterized cre driver lines, but having said that, there has been some drift with a number of these lines (for example the nestin-cre mouse used here), and the authors really should provide their own data to validate the mutants they are studying.* We have included cre control data as Figure 1—figure supplement 2

*3) The section on neuronal mislocalization is very interesting, but needs to be strengthened in a number of additional ways. First, the authors need to provide some kind of quantification, focusing on the E545K mutants, to solidify their data. This could be as simple as dividing the cortex into zones/layers (depending upon the age), and quantifying the proportion of layer-specific marker-positive cells that are in the different zones/layers in controls versus mutants.*

We provide extensive new quantitation in Figure 6.

*Second, the authors should remove their references to neuronal migration. They don't actually study migration anywhere in this manuscript, but instead study localization of neurons. Since, as they indicate in the discussion, these phenotypes could be due to many different developmental perturbations, to indicate that they are due to neuronal migration deficits is misleading.* Our new birthdating experiments (Figure 6) now provide extensive support for migration deficits.

*4) The approach of RPPA to scan a large number of signaling proteins at once is a valuable one. Nonetheless, it is important that these changes be validated using other approaches. In particular, the authors should choose at least a few of the more important changes and verify them using western blots. If they all match the data obtained using RPPA, then this will validate the entire approach for these particular samples.*

The amount of mutant tissue available from the brain of the mice was limiting, particularly given the different assays that were performed on the tissues. RPPA requires very small amounts of tissue to perform multiplex analysis across many antibodies and was thus the most tissue sparing approach available. Several studies have demonstrated that RPPA is more sensitive than standard Western blotting and is becoming a standard assay in cancer biology (Akbani et al., 2014; Tibes et al., 2006; Gujral et al., 2013; Hennessy et al., 2010; Nishizuka et al., 2003). Indeed the RPPA facility has used this specific panel on more than 100,000 samples (Gordon Mills, our co-author, runs this facility and has published extensively on RPPA). Well over 100 papers including almost all of the TCGA (The Cancer Genome Atlas project) papers have used RPPA analysis and indeed, no confirmation western blots have been requested for the complete TCGA set due to the extensive validation in other sample sets. The three independent biological replicates per sample further support data validity. Unfortunately, insufficient mutant brain tissue was available to perform western blots at the end of the study and thus we had to rely on the results from the RPPA analysis.

Reviewer #2:

*In this study the authors create animal models of megalencephaly (MEG) that are associated with mutations in PIK3CA. They first show that overexpression of PIK3CA in neural progenitors disrupts normal cortical development, and demonstrate that the timing of transgene activation is a critical feature of disease severity. They also demonstrate that overexpression of the two PIK3CA alleles leads to increased size of the cortex that is generally associated with an increase in nuclear and cell size and a decrease in cell density. They then demonstrate that the expanded cortex displays defects in lamination. As a consequence, transgenic animals overexpressing active alleles of PIK3CA are more prone to epileptic seizures. Finally, the authors show that a class I PI3K inhibitor has anti-seizure effects, and reduces phosphorylation of PI3K pathway components.*

*1) Validation of animal models. The authors need to show exactly when and where the p110a transgenes are expressed by their nestin-cre, nestin-creER and hGFAP-cre driver lines. Could there be differences in phenotypes from these different cre drivers based on level of expression, rather than simply due to the timing of cre activation? Also, the timing of tamoxifen administration for nestin-creER should be given in the results section to know when the transgenes were induced with this driver.*

Cre validation is now provided in Figure 1—figure supplement 2.

Figure 1 clearly shows that it is not the cre line that affects the phenotype per se. It is the nature of the specific mutant allele as well as the time of activation of the mutation in brain that result in the developmental phenotypes. For example, if we compare the *E545K* and *H1047R* mutants, both activated by *hGFAP-cre*, we observe that the *hGFAP-cre;H1047R* mutant has hydrocephalus and agenesis of corpus callosum, while neither of these phenotypes are observed in *hGFAP-cre;E545K* mutant.

*2) Comparisons of animal models. The authors do a nice job of presenting the different phenotypes associated with overexpression of the E545K allele achieved with the different cre drivers, but it would help to have more information on the H1047R allele. How does overexpression of H1047R affect brain size and morphology with the Nestin-creER and nestin-cre lines? The authors stated that the most severe phenotype was achieved in hGFAP-cre;H1047R mtuants, but relative to what? To the other H1047R lines, or relative to E545K lines (or both)? If H1047R was only tested with the hGFAP-cre line, this should be clearly stated.*

We have rephrased this in the text. We meant among all the mutants studied in this paper, irrespective of allele or cre line, *hGFAP-cre;H1047R* mutants show the most severe developmental defects in the brain.

Reviewer #3:

*This manuscript describes the phenotype of mouse models with mutations in PIK3CA that should overactivate the PI3K-AKT pathway in the nervous system. The E545K and H1047R activating mutations have been found before in patients with cancer, but also with various forms of segmental cortical dysplasia. Corresponding mouse lines have been generated and investigated before for studying the role of these activating mutations in cancer. In this study, the authors have used Nestin-Cre and GFAP-Cre for nervous system-specific induction of the mutant alleles, and these mice developed enlarged brain size, cortical dysplasia and epilepsy. The authors find that GFAP-Cre H1047R mutants show a very severe phenotype with progressive hydrocephalus and death around birth. Mice with E545K mutation and the same GFAP-Cre driver were less affected. Earlier developmental induction of the E545K mutation with Nestin-Cre showed a more severe phenotype than that observed after GFAP-Cre-mediated activation. The authors then found that the inner cortical length of GFAP-Cre H1047R mutants at E14-E16 was longer than in controls. Surprisingly, increased proliferation was not observed in these animals, only enlarged cell size. Interestingly, the Nestin-Cre E545K mutants showed enhanced proliferation with reduced cortical thickness, indicating that the newborn cells did not adequately migrate. The authors also observed dysplastic Cajal-Retzius cells in GFAP-Cre H1047R mice at E16. They then investigated Nestin-Cre and Tamoxifen-inducible Nestin-Cre E545K mutant mice for abnormalities in EEG and found signs of epileptiform activity, in particular when induced with the GABA-A receptor antagonist PTZ. The authors then also provide evidence that the PI3K inhibitor BKM-120 suppresses seizure activity and shows some effects on parameters of PI3K pathway activation. Although these data are of potential interest to understand the molecular basis of segmental cortical dysplasia in humans, the paper is largely descriptive and the mechanisms how PI3K activation induces this complex disorganization of cortical development that then results in altered excitability and epileptic seizures is not explained. It is even unclear whether alterations in cell division, cell cycle exit, migration, terminal differentiation, spontaneous activity and synaptic plasticity occur in these mice.*

We concur that we have extensively described the first models of human *PIK3CA* related brain malformations – a clinically important class of cortical malformations with an impressive phenotypic spectrum – from dysplastic megalencephaly to focal cortical dysplasia. Our mouse models recapitulate the human phenotypes and demonstrate validity of our models. Further we demonstrate that the mice display epilepsy – perhaps the most clinically relevant phenotype in this population. Since these are the first models, extensive phenotypic description was required.

Mechanistically, we counter that we have indeed defined the developmental basis of megalencephaly and cortical dysplasia, showing alterations in proliferation, cell cycle exit, cell migration versus cell fate, and cell size (Figure 3, Figure 6). While these phenotypes are important, the major significant finding of our work is that the cortical dysplasia is less relevant to epilepsy than previously thought. Developmentally, we can dissociate dysmorphology from epilepsy. Mice with abnormal neocortical cell distribution (*Nestin-cre;E545K*) as well as those without dysplasia (*Nestin-creER;E545K*) exhibit epilepsy. Further, we can acutely treat the epilepsy in mice even with severe dysplasia.

As we have now more lucidly argued in the Introduction and Discussion, the finding that Pik3ca-driven epilepsy is an active process, amenable to acute intervention is a major advance. Mechanistically, acute treatment with BKM120 alters mTOR-dependent phosphorylation of AKT with strong effects in the mutant hippocampus. Given the extensive developmental analyses presented in addition to the epilepsy data, we consider further analysis of the synaptic physiology and the dissection of pAKT-dependent and/or pAKT-independent pathways that underlie the epilepsy to be well beyond the scope of the current manuscript.

*It is unclear whether GFAP-Cre and Nestin-Cre act in the same or in distinct precursor cell populations, whether defects in all types of neurons, or only specific subtypes, or defects in astrocytes cause this specific phenotype. The latter is possible in particular in GFAP-Cre mice.*

Please see Figure 1—figure supplement 2 for differential expression pattern of *hGFAP-cre* and *Nestin-cre* lines. These two lines express in overlapping but distinct cell populations. Moreover, when we compare *Nestin-cre;E545K* and *hGFAP-cre;E545K* we see they have comparable morphological phenotypes while *hGFAP-cre;H1047R* is quite different. This confirms that differential phenotype is dependent on the specific mutant allele even when same cre line is used.

*The previous paper by Robinson et al. in Nature 2012 showed that the same E545K mutation, when induced with Blbp-Cre in lower rhombic lip progenitors accelerates tumor generation, but does not per se initiate WNT subgroup medulloblastoma. This leaves the question open whether the same mutation in Nestin- or GFAP-Cre expressing cells causes defects that need additional signals to develop the pathogenic phenotype.*

We are unclear regarding the nature of the reviewer’s concern. We demonstrate that *E545K* mutation driven by either *Nestin-cre* or *hGFAP-cre*, result in megalencephaly, cortical and white matter dysplasia. The nature and extent of dysplasia depends on the developmental timing. We have also provided two genetic models of epilepsy caused by the *E545K* mutation; and we show seizure activity can be acutely suppressed by a pan-Class I PI3K inhibitor, BKM120. Moreover, we provide RPPA protein analyses which show that the protein levels are dynamically regulated by acute administration of the drug. These data conclusively show that activation of PIK3ca *alone* is sufficient to give rise to different developmental defects including epilepsy.

*Additional specific comments: 1) The paper lacks clear evidence about the types of cells in which the mutant PIK3CA isoforms are expressed, and at which stage of development recombination occurs. This is necessary to explain whether dysregulated PIK3CA causes abnormalities in division of neural precursor cells.*

We have included cre control data as Figure 1—figure supplement 2.

*I wonder why the authors did not do more detailed pulse chase experiments with BrdU to study potential effects on altered proliferation of different types of precursor cells in the ventricular and subventricular zone. It would also be important to know whether proliferation of neuronal cells is differentially affected in comparison to proliferation of astrocytic precursor cells. At E16.5, the time point when the analyses shown in Figure 2 were performed, proliferation of astrocytes has already started, and the authors should do the studies shown in Figure 3 with markers that can distinguish neuronal from astrocytic precursor cells.*

We have now included detailed BrdU-pulse experiments to study proliferation (1 hour pulse), cell cycle exit (1 day pulse) and birthdating early and late neurogenesis (Figure 3 and Figure 6). We have also quantitated the differential BrdU distribution in control and mutant littermates in different zones of the cortical column. We have qualitatively shown that the *Pik3ca E545K* mutation does not affect astrocyte shape, size or number (Figure 5—figure supplement 3). More detailed astrocyte analysis was deemed beyond the scope of the current study.

*2) It is hard to explain why the cell density, as shown in Figure 3 is reduced when BrdU incorperation is increased. It is also not clear in which layers of the cortex the authors observe this reduced cell density at E16.5. To understand this phenotype, the VZ/SVZ, the mantle zone and the intermediate zone should be investigated separately.*

In E16.5 *Nestin-cre;E545K*, multiple cellular mechanisms collectively affect the final phenotype. E16.5 *E545K* mutant has increased labeling index (indicating increased proliferation, Figure 3) as well as increased cortical length (Figure 2). The cortical density at E16.5 is not distinctly lower in any particular zone. Additionally at a later developmental time-point P0, we quantitate number of cells in different zones of the cortical column, as described in Figure 6.

*Did the authors also investigate whether there is enhanced loss of cells by apoptosis or other types of cell death between E14.5 and E16.5 to explain this phenotype?*

See reply to Reviewer #2’s comment 6: In Figure 3—figure supplement 1 our mutants actually show slight decreases in cell death during embryonic peak neurogenesis.

*3) The phenotype with increased cell size is hard to explain. This could be due to a defect in the cell fate determination of distinct populations of neuronal types, for example when generation and survival of pyramidal cells in comparison to interneurons is enhanced, or also by altered generations of neurons in comparison to glial cells. In order to understand the phenotype of altered cortical morphology and enhanced epileptic seizures, this would be important to know.*

mTOR overactivation is known to cause increased cell size (Blumcke et al., 2011; Fingar et al., 2002; Lim et al., 2015; Ljungberg et al., 2006). As we have indicated in the Discussion, in human *PIK3CA* brains and in cultured rodent hippocampal neurons overexpressing Pik3ca, cell size is increased, but not to the extent seen with TSC mutation, PTEN deletion, or pAKT activation.

We see no evidence of changes in cell fate (Figure 6). There is a temporal dependence on cell size, with postnatal *Pik3ca* activation having no effect on cell size (Figure 3—figure supplement 2). Thus, altered cell size cannot be a common basis of epilepsy in our models as both models exhibit epilepsy.

Interneurons are born at embryonic stages and their migration is almost complete before birth. Postnatal activation of *Nestin-creER;E545K* does not cause increased cell size or megalencephaly. The YFP-positive cells marking the cells in this model that express mutant PiK3ca do not coexpress interneuron markers like Parvalbumin and Somatostatin (data not shown). Altered interneuron development in mice where mutant *Pik3ca* is activated embryonically (*Nestin-cre;E545K*) may contribute to dysmorphology. However, this does not seem to be the common mechanism behind epilepsy since both embryonic and postnatal *Pik3ca* activation causes epilepsy.

*4) There is a big gap between the defects in cortical thickness and the potential mechanisms that could explain the altered threshold for epileptic seizures. Is there a dysbalance between GABAergic and glutamatergic neurons and synapses in the brain of these mice? Is there defective regulation of synapse generation, function and plasticity? Are there defects in dendritic arborization and synaptogenesis? Are there differences in the morphology of astrocytes and differences in gene expression in such astrocytes, i.e. for neurotransmitter transporters, that could explain the epileptiform activity?*

Although we have extensively characterized the cortical dysmorpholgy in our models, our observation that the epilepsy is acutely sensitive to treatment *despite dysmorphology* argues that it is an active Pik3ca-dependent process. This is a major and paradigm shifting finding. Given the extensive developmental analyses presented in addition to the epilepsy data, we consider further analysis of the synaptic physiology and the dissection of pAKT-dependent and/or pAKT-independent pathways that underlie the epilepsy to be well beyond the scope of the current manuscript; however they are the focus of ongoing studies in our lab.

5) It is not clear whether the enhanced epileptiform activity reflects a developmental defect or altered synaptic activity, which is directly driven by altered PIK3CA activation. The authors have used both a constitutive Nestin-Cre and a Tamoxifen-inducible Cre driver, but there are no data to distinguish the phenotype when E545K mutant PIK3CA is expressed from developmental stages, or only after acute induction in the adult.

While developmental defects may contribute to Pik3ca-driven epilepsy, our combined data from multiple models argue that the developmental defects are not the major drivers. Morphological differences between embryonic and postnatal E545K activation are summarized in Figure 1 and Table 1.

*Other questions and points of concern: 1) In the methods section, the authors say that all mouse lines were maintained on a mixed genetic background. It is not clear what this means and whether differences in genetic background could influence the phenotype. How can the authors exclude the possibility that the more severe phenotype observed in the GFAP-Cre drivers in comparison to Nestin-Cre drivers is due to different genetic background?*

We have rewritten the Materials and methods section to clarify the nature of the mixed background. Data presented in Figure 1 demonstrates that the genetic background of mice does not broadly affect Pik3ca-related phenotypes.

*2) The data shown in Figure 5 are very hard to understand and not well described. What is the function of time at which the numbers of seizures occur? Does the diagram mean seizures within one minute? These bars lack standard deviations. How can one understand i.e. 0.4 seizures or 1.6 seizures per minute and why does the number of seizures fluctuate with a peak at 7 min., 15 and 22 min. after PTZ injection? Which phase of seizure activity is affected by BKM-120-treatment? Obviously, the PI3K inhibitor does not reduce seizure activity completely.*

We have clarified the figure in the text and figure legend. During the 30 min after PTZ injection, mice experience several seizures. For each mouse we counted the number of seizures that they experience in one min bins to capture the complete seizure burden during our experiments. Therefore, the histogram is a representation of the average number of seizures that mice within control and mutant groups experience post-PTZ administration in 1 minute bins. These are population averages, with periodic fluctuations reflecting postictal refractory periodicity (Lado et al., 2008). Overall statistical analyses are presented in Figure 7. BKM120 reduces total number of seizures, seizure severity and increases latency post-PTZ administration. BKM120 normalizes PTZ-induced seizures to untreated control (non-mutant) values.

3) The data shown in Figure 6 are hard to understand, given that the differences are generally much lower than expected, and it is also not clear which types of cells and which subpopulations of neurons are predominantly affected by altered Akt phosphorylation or other downstream signaling parameters that are expected to be altered when the PI3-K pathway is constitutively activated.

PI3K signaling is highly regulated through feedback and feed-forward loops. As we have stated in the Discussion, our proteomics data is congruent with published breast cancer data showing that even high levels PI3K pathway overactivation only causes modest changes in the levels of steady-state PI3K signaling. Our cre validation data shows cell populations predominantly affected in each mutant.

References:

R. Akbani et al., Mol Cell Proteomics 13, 1625 (2014).

T. S. Gujral et al., Oncogene 32, 3470 (2013).

B. T. Hennessy et al., Clin Proteomics 6, 129 (2010).

S. Nishizuka et al., Proc Natl Acad Sci U S A 100, 14229 (2003).

D. C. Fingar, S. Salama, C. Tsou, E. Harlow, J. Blenis, Genes Dev 16, 1472 (2002).

M. C. Ljungberg et al., Ann Neurol 60, 420 (2006).

F. A. Lado, S. L. Moshe, Epilepsia 49, 1651 (2008).